

# Tailor-made spatial patterns for hydrological model parameters combining regionalisation methods

Laura Rouhier[1,2], Federico Garavaglia[1], Matthieu Le Lay[1], Timothée Michon[3], William Castaings[3], Nicolas Le Moine[2], Frédéric Hendrickx[4], Céline Monteil[4], and Pierre Ribstein[2]

[1]Électricité de France, DTG, Grenoble, France
[2]Sorbonne Université, Paris VI, UMR 7619 METIS, Paris, France
[3]TENEVIA, Meylan, France
[4]Électricité de France, R&D, Paris, France

**Correspondence:** Laura Rouhier (laura.rouhier@edf.fr)

**Abstract.** Calibration of spatially distributed models is an important issue given their over-parameterisation. Three common regionalisation methods can be distinguished: transposition, prescription and constraint. This paper proposes a strategy where the three methods are combined to provide several spatial patterns depending on the model parameters. On the one hand, insensitive and equifinal parameters are prescribed uniformly while parameters with a physical meaning are prescribed at the

5 mesh scale. On the other hand, parameters linked with a proxy runoff signature are constrained over each sub-basin and the remaining parameters are transposed with a physio-climatic pattern constructed over the calibration sub-basins.

The above tailor-made pattern regionalisation is applied over two large French catchments, the Loire catchment at Gien (35,707 km$^2$) and the Durance catchment at Cadarache (11,738 km$^2$), at the daily time step. It is then evaluated and compared to a single regionalisation method over dozens of validation stations, treated as ungauged during the parameter regionalisation

process. To do so, observed and simulated streamflows are compared in light of 4 runoff signatures: daily runoff, seasonality, flood and low flow. The results show that the tailor-made patterns succeed in significantly enhancing almost all the signatures. The enhancement appears for the least well-modelled stations and contributes a 20% improvement towards gauged modelling. It also tends to guarantee a minimum performance in an ungauged context since the minimum KGE is now 0.4 whatever runoff signature is used.

## 1 Introduction

Spatially distributed hydrological models allow for (i) spatially distributed climatic inputs, (ii) spatially distributed model parameters, (iii) ungauged simulations and (iv) upstream-downstream consistency. With the increasing availability of spatial data and the improvements in computational power, this type of model represents a real potential for hydrological modelling. The Distributed Model Intercomparison Project (Smith et al., 2004; Reed et al., 2004; Smith et al., 2012, 2013) investigated

the capabilities of existing distributed hydrologic models. However, this project did not provide any recommendations on parameter estimation schemes. The strategy is not as well defined as for lumped models whose parameters usually follow from calibration over the observed outlet streamflow. Indeed, in distributed models, each spatial unit comprises one set of parameters





while most of these units are ungauged (Sivapalan et al., 2003; Hrachowitz et al., 2013). Distributed models therefore suffer from over-parameterisation and equifinality (Beven and Hornberger, 1982; Beven, 2001). To overcome these difficulties, one can make use of three regionalisation methods: (i) transposition, (ii) prescription and (iii) constraint.

Transposition consists in grouping the $N_u$ spatial units into $N_r$ regions, each of them comprising one set of $N_p$ parameters calibrated over gauged discharge stations. This method reduces the dimensionality of the optimisation problem from $N_u \times N_p$ to $N_r \times N_p$ free parameters. The region delineation can follow from physio-climatic similarity (Beldring et al., 2003; Kumar et al., 2013) or a gauged network (Andersen et al., 2001; Feyen et al., 2008; Khakbaz et al., 2012; De Lavenne et al., 2016). With this method, Andersen et al. (2001) and Feyen et al. (2008) succeeded in significantly enhancing the performance, moving from a uniform parameter model to a spatialised parameter model.

Prescription is based on *a priori* or empirical relationships between model parameters and catchment characteristics such as soil properties (Koren et al., 2000; Twarakavi et al., 2009). That way, Andersen et al. (2001) and Khakbaz et al. (2012) tested an uncalibrated model with distributed parameters directly estimated from field data, the literature and previous studies. However, within the framework of distributed modelling, prescription is quite often enhanced with transposition to reduce the gap between modelling and the physical assessment (Francés et al., 2007; Smith et al., 2013). Francés et al. (2007); Pokhrel and Gupta (2010); Samaniego et al. (2010) and Khakbaz et al. (2012) first prescribed spatial parameter fields from catchment characteristics such as soil information, land cover, vegetation and topography. Then, they adjusted them through transposition of uniform correction coefficients (*i.e.* calibration of one region) called superparameters or global parameters calibrated over the observed outlet streamflow. For instance, Pokhrel and Gupta (2010) defined three superparameters per model parameter: a multiplying, an additive and a power coefficients involving the calibration of $3 \times N_p$ superparameters. With this approach, they succeeded in considerably improving the basin outlet performance compared to the single prescription method, even if it was insufficient to enhance modelling of interior pseudo-ungauged points. Prescription and transposition can also be inverted by first calibrating the model parameters uniformly and then modifying them with spatial fields estimated from catchment characteristics without further calibration (Koren et al., 2004; Khakbaz et al., 2012). Khakbaz et al. (2012) then obtained better results at three interior pseudo-ungauged points in comparison to prescription before transposition. To a more limited extent, prescription can also be a tool to calibrate the model parameters to be transposed. Similar to Ajami et al. (2004), every region of the catchment can be calibrated over its observed outlet streamflow by temporarily prescribing the downstream parameters with catchment characteristics.

Finally, constraint is based on proxy data, *i.e.* on the hydrological signature estimated without any streamflow measurement that can give a clue about the catchment's hydrological behaviour. Constraint consists in using these proxy data on top of or even instead of streamflow time series as a constraint in the calibration process. With the increasing number of available data, this method is largely addressed in the literature. We mention a few of these studies here. Madsen (2003) introduced a multi-objective calibration framework where a distributed model is calibrated over both observed outlet streamflow and groundwater levels measured at 17 interior wells. Instead of groundwater data, Khan et al. (2011) and Silvestro et al. (2015) suggested calibrating the distributed model parameters with remote-sensing data. Along with streamflow observations, Khan et al. (2011) used satellite-derived flood maps to calibrate a module of a distributed model designed for flooding. Their results proved to be





very promising for distributed hydrological model calibration, even in ungauged basins or data-sparse regions. Silvestro et al. (2015) then highlighted the usefulness of land-surface temperature and surface soil moisture satellite observations to reduce parameter equifinality. Their results also confirm that the constraint method is a convenient alternative to calibrate a model in an ungauged context since they ended up with similar model performance for calibration over solely satellite-derived data and

solely streamflow observations.

The aim of this paper is to go a little further by combining the three above-cited regionalisation methods. The model parameters are spatialised with one of the three methods according to their characteristics and hydrological meaning. Four parameter spatial patterns follow from this multi-faceted method: a uniform pattern, a hydrological mesh pattern and two intermediate patterns. The method is applied over two French mesoscale catchments, the Loire at Gien and the Durance at

Cadarache, for the 1980-2012 period. Using a 50/50 spatial split-sample test, the performance of the tailor-made patterns is assessed over pseudo-ungauged stations and compared with that of a single transposition scheme.

Section 2 introduces the spatially distributed hydrological model along wih the evaluation criteria. Section 3 presents the data set. Section 4 describes the parameter spatial patterns and section 5 shows the results. Finally, section 6 gives some conclusions and perspectives.

## 2 Modelling

### 2.1 The distributed hydrological model

The hydrological model used in this work is the spatially distributed conceptual model named MORDOR-TS, presented in Rouhier et al. (2017). In this model, the basin is split into hydrological meshes, themselves vertically divided into elevation zones. The hydrological meshes are connected to each other through the hydrographic network, each of them having its own set

of parameters. At every time step, (i) the runoff production is calculated independently over each mesh, (ii) then all production is routed from mesh to mesh to the simulation points.

The production module quantifies the exchanges between six interconnected store components. Using precipitation and air temperature time series, they fill up and drain to supply the rivers as illustrated in Fig. 1. The discretisation into elevation zones, designed for mountainous regions, is only activated for the Durance catchment. For further detail about the production

processes, readers may refer to Garavaglia et al. (2017).

The runoff contribution estimated for each hydrological mesh is then propagated to the simulation points through the hydrographic network as described in Fig. 2. The routing module achieves the intra-mesh and inter-mesh propagation. It consists of a 1D diffusive wave model, for which the parameters of celerity and diffusion are independent of streamflow, as developed by (Hayami, 1951; Litrico and Georges, 1999). The MORDOR-TS model is governed by 22 free parameters in its entire for-

mulation. However, the model is ajustable depending on the basin's characteristics and complexity. In this study, the version comprises 12 free parameters for the Loire catchment and 16 for the nival Durance catchment. Details about the parameters are given in A.




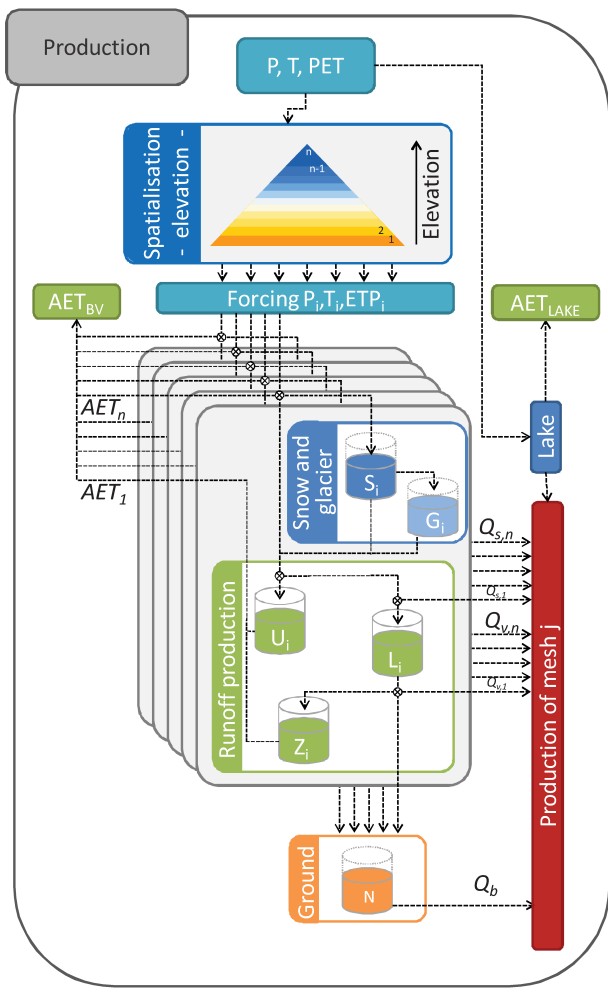

**Figure 1.** Overview of the production module which is applied independently to each hydrological mesh of the MORDOR-TS distributed model.

## 2.2 Calibration and evaluation numerical criteria

The hydrological model is expected to provide a sound hydrological behaviour. Consequently, it must faithfully reproduce the various hydrologic dynamics, summarised by several streamflow signatures (Blöschl et al., 2013). To do so, the simulated and observed streamflows are compared in the light of 4 numerical criteria based on the Kling-Gupta Efficiency (KGE, Gupta et al.

5 (2009)) applied to four following runoff signatures (Garavaglia et al., 2017; Rouhier et al., 2017):

- the entire streamflow time series (KGE daily runoff), which results from the combination of all the processes,

- the long-term mean daily streamflow (KGE seasonality), which focuses on the capacity to reproduce seasonal variation of observations,





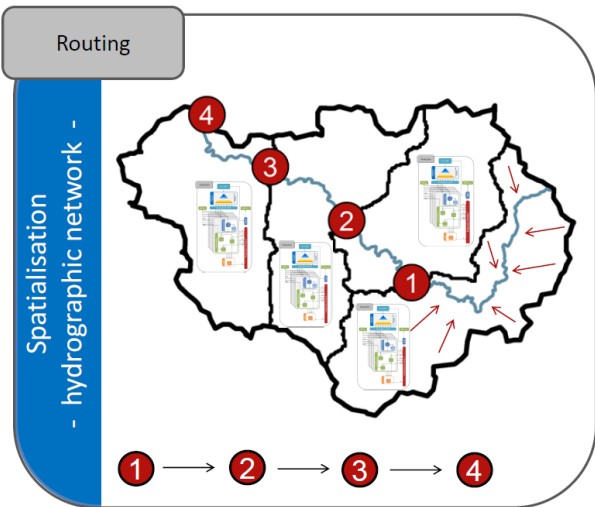

**Figure 2.** Overview of the routing scheme of the MORDOR-TS distributed model: intra-mesh and inter-mesh propagation.

- – the flow duration curve (KGE flood), which focuses on the capacity to reproduce high flow generated by highly dynamic interactions,

- – the flow recessions during low flow periods (KGE low flow), due to long-term processes.

These four KGE criteria are used for both calibration and spatial evaluation.

For parameter calibration, the four criteria are implemented in the multi-objective genetic algorithm caRamel[1] (Monteil et al., submitted). As part of the constraint method, an additional criterion is used as a fifth objective function as introduced in section 4.2.3. For model warm-up, a systematic first 1-year period is used. After about 5000 model runs, a 4D Pareto frontier is provided by the algorithm (Yapo et al., 1998). An optimum set is then selected within the Pareto frontier by calculating the Euclidian distance on ranks.

# 3 Data set

## 3.1 Study area

The tailor-made method is evaluated over two wide French catchments: the Loire catchment at Gien (Fig. 3a) and the Durance catchment at Cadarache (Fig. 3b). The first one is located in the center of France and covers 35,707 km$^2$, . Its elevation varies between 118 and 1838 m.a.s.l., the peak of the summits of the Massif Central. It is a mainly pluvial catchment with a median elevation of 417 m.a.s.l. The Durance basin at Cadarache, covering 11,738 km$^2$, is located in the Alps in south-east part of

[1]https://cran.r-project.org/web/packages/caRamel/index.html



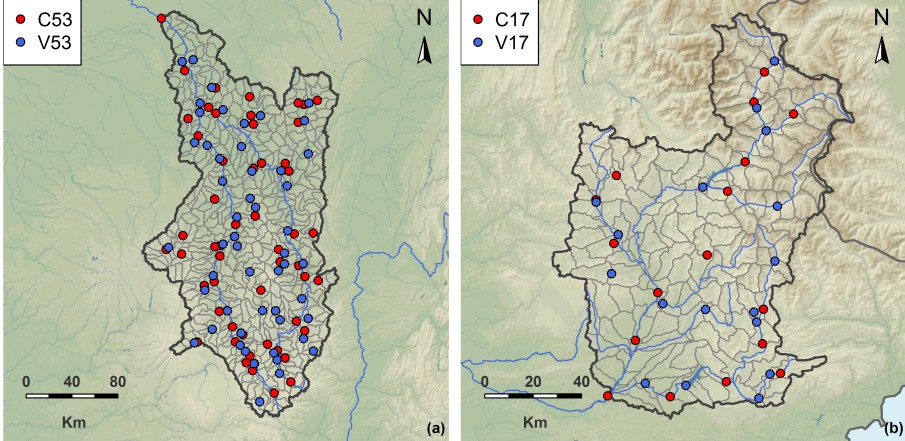

**Figure 3.** The two catchments studied: (a) the Loire catchment at Gien with its 106 discharge stations and (b) the Durance catchment at Cadarache with its 34 discharge stations. The red dots indicate the stream gauges belonging to the calibration sample, while the blue dots indicate those belonging to the validation sample. The grey units represent the hydrological meshes.

France. Its elevation varies between 247 and 4102 m.a.s.l., the peak of the Barre des Écrins. With 60% of the basin above 1000 m.a.s.l., the upper part is nival while the lower part is nivo-pluvial.

For distributed purposes, the Loire and the Durance catchments are discretised into 387 and 133 hydrological meshes, respectively. They are indicated by the grey units in Fig. 3. These hydrological mesh patterns arise from a 100-m DEM with a mesh target area of 100-km$^2$.

## 3.2 Climatic input data

Climatic inputs are obtained by a method similar to SPAZM (Gottardi et al., 2012). They follow from a statistical reanalysis of the ground network, based on reliefs and weather patterns (Garavaglia et al., 2010). The precipitation and air temperature fields over the Loire catchment at Gien (resp., Durance catchment at Cadarache), are built from 146 (resp., 115) rain gauges and more than 100 (resp., more than 70) temperature gauges. They are available at 1-km$^2$ and 1-day resolution between 1948 and 2012. In the herein study, we only used the data from 1 September 1980 to 31 August 2012.

## 3.3 Streamflow data

Daily streamflow time series are collected from the databases compiled by French national environmental agencies and Électricité de France (EDF). Their selection is based on observation time availability over the 1980-2012 period, drainage area, as well as quality and temporal homogeneity of the time series. Within the Loire catchment, the 106 time series selected, the observation period ranges between 7 and 32 hydrologic years, with an average of about 22 years per station. The stations are located in Fig. 3a (red and blue dots). Their drainage areas vary between 100 and 35,707 km$^2$ with an average of 2,844 km$^2$. Inter-annual runoff ranges from 136 to 1057 mm per year. Among the 106 basins, the inter-annual precipitation ranges from




729 to 1495 mm per year. The inter-annual ratio between precipitation and potential evapotranspiration $\frac{P}{PET}$ (Andréassian and Perrin, 2012), named humidity index, varies between 1 and 2.7 with an average of 1.5.

Within the Durance catchment, 34 time series were selected. In Fig. 3b, their locations are represented by the red and blue dots. The observation period of the 34 time series ranges between 6 and 32 hydrologic years, with an average of about 22 years per station. Their drainage areas vary between 94 and 11,738 km$^2$ with an average of 1,275 km$^2$. Among the 34 catchments, runoff ranges from 162 to 881 mm/year and precipitation from 898 to 1321 mm/year. The Durance catchment is more humid than the Loire with a mean humidity index of 2, ranging from 1.2 to 3.

### 3.4 Spatial split-sample test

The parameterisation of the distributed model is assessed over pseudo-ungauged stations through a 50/50 spatial split-sample test introduced in Rouhier et al. (2017). Over each catchment, the discharge data are divided into two similar subsets: a calibration and a validation station sample. The two samples are equal in the number of stations, spatially homogeneous and as similar as possible in terms of temporal, climatic and physio-graphic characteristics. Hereafter, the calibration and the validation samples of the Loire catchment comprising 53 gauges each are referred to as C53 and V53. Similarly, those of the 17-gauge Durance catchment each are referred to as C17 and V17. They are presented in Fig. 3. The red dots represent the calibration stations and the blue dots the validation stations. The calibration stream gauges are the gauged stations whose streamflows are used to calibrate the parameters. In contrast, the validation stations are pseudo-ungauged stations: their streamflow time series are never used to calibrate or even estimate the model parameters but are used *a posteriori* to evaluate the parameter regionalisation.

Our validation scheme is not exclusively spatial. In fact, it lies between spatial and spatiotemporal depending on the periods' intersection (Patil and Stieglitz, 2015), since the runoff observation periods of the calibration and validation stations are not necessarily identical.

### 4 Parameter spatial patterns

### 4.1 From a single sub-basin pattern...

First, the model parameters were all spatialised at the same resolution: a spatial pattern based on the calibration sub-basins, introduced in Rouhier et al. (2017). This pattern consists in dividing the catchment into sub-catchments delineated by the upstream and downstream calibration stations, as previously done by Feyen et al. (2008) and De Lavenne et al. (2016) Every sub-basin is associated with one set of parameters calibrated on its outlet station with the upstream streamflow injected at the upstream nested calibration stations. Figure 4a presents the pattern obtained over the Loire catchment while Fig. 5a gives the pattern obtained over the Durance catchment. Each colour represents one calibration sub-basin, *i.e.* one set of parameters.





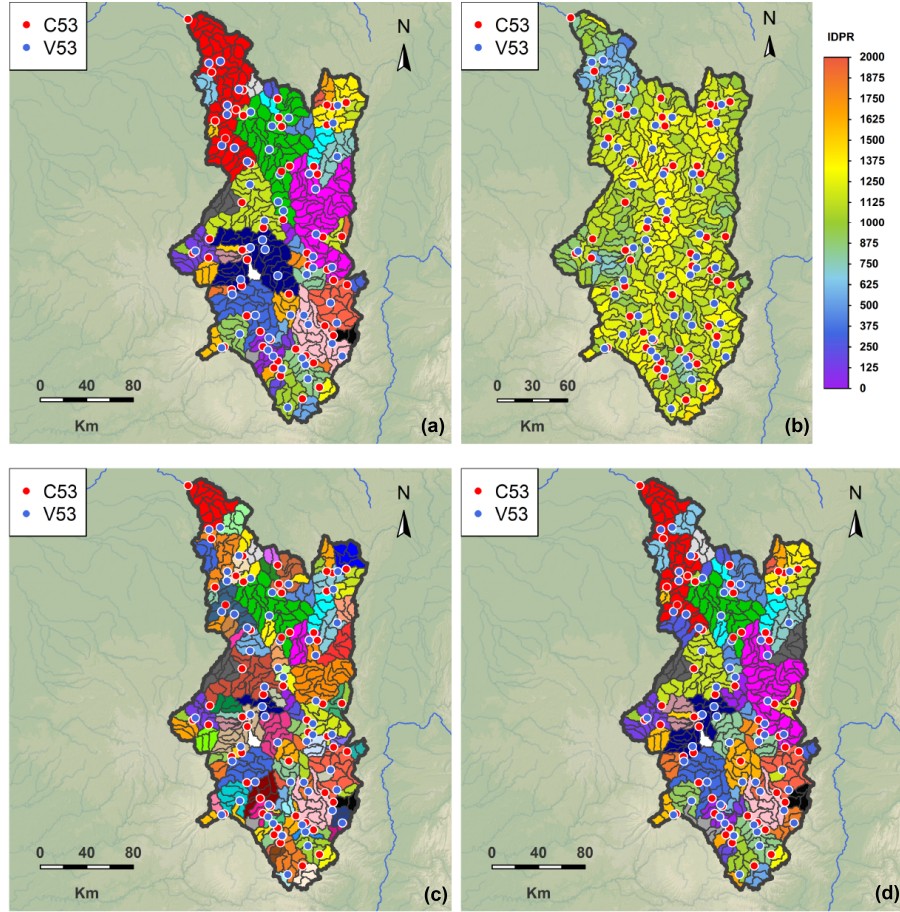

**Figure 4.** Parameter spatial patterns of the Loire basin at Gien. Each colour stands for one set of parameters. The several patterns correspond to several degrees of spatialisation which are: (a) the initial sub-basin pattern, (b) the hydrological mesh pattern, (c) the sub-basin pattern and (d) the physio-climatic sub-basin pattern. The parameters are assigned to the spatial patterns according to their characteristics.

## 4.2 ... to tailor-made patterns

The objective was then to improve the spatialisation by adapting the spatial pattern and the regionalisation method to several parameters. To do so, we conducted an incremental experimental framework based on trial and error. This framework, although not exhaustive, is based on feedback from the model and on an exhaustive sensitivity analysis of the model's parameters. After dozens of experiments, we propose four spatial patterns combined with the three regionalisation methods, as shown in Fig. 6. These four spatial patterns are detailed in the following sections.





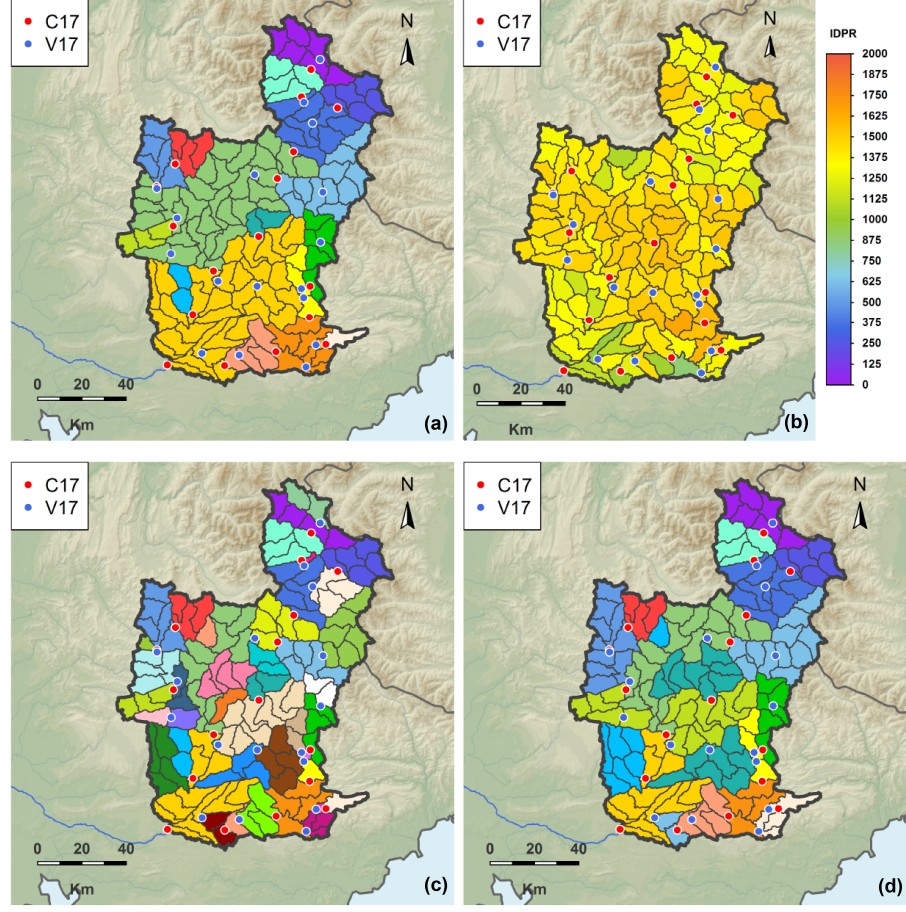

**Figure 5.** Parameter spatial patterns of the Durance basin at Cadarache. Each colour stands for one set of parameters. The several patterns correspond to several degrees of spatialisation which are: (a) the initial sub-basin pattern, (b) the hydrological mesh pattern, (c) the sub-basin pattern and (d) the physio-climatic sub-basin pattern. The parameters are assigned to the spatial patterns according to their characteristics.

### 4.2.1 The uniform pattern for insensitive or equifinal parameters derived from prescription

A sensitivity analysis of the MORDOR-TS model has been conducted over several French catchments according to the approaches of Sobol (1993), Homma and Saltelli (1996) and Liu and Owen (2006). The analysis informed us that five parameters were not sensitive at all over the Loire catchment: the parameter generating the delayed flows ($evl$), the snow parameters ($kf$ and $lts$), the distribution parameter between the root zone and the capillarity storage ($kzu$) and the diffusivity of the diffusive wave transfer function ($dif$). Over the Durance catchment, the first three parameters were significantly sensitive while the distribution parameter ($kzu$) and diffusivity ($dif$) proved to be insensitive. Consequently, we propose to prescribe the insensitive parameters uniformly at their default value, namely five parameters for the Loire catchment and two for the Durance catchment.







**Figure 6.** Overview of the tailor-made method, which combines three regionalisation methods to provide four spatial patterns according to the characteristics of the model parameters.

Moreover, the sensitivity analysis pointed out the equifinality between the two parameters governing the draining of the groundwater reservoir ($evn$ and $lkn$). One of the two is thus prescribed uniformly to its default value while the other is left free.

### 4.2.2 The hydrological mesh pattern for sensitive parameters derived from prescription

5    In the hydrological model, potential evapotranspiration (PET) is affected by vegetation through a crop coefficient formulation (Allen et al., 1998; Allen, 2003). Crop evapotranspiration (CET) is estimated as CET = PET $\times$ K$_C$ where K$_C$ stands for the crop coefficient. According to Hunsaker et al. (2003), Duchemin et al. (2006), Rafn et al. (2008) and Mutiibwa and Irmak (2013), the crop coefficient is linked with the Normalized Difference Vegetation Index (NDVI) derived from satellite observations (Solano et al., 2010). We can therefore implement it as an appraisal of the crop coefficient time series of the model which was

10    previously obtained through a one-parameter formulation where the parameter was set uniformly to its default value. To do so, the inter-annual time series of NDVI (16-day and 1-km$^2$ resolution) is aggregated at the mesh scale and then used to prescribe K$_C$ at the same scale.

   In the hydrological model, another parameter stands for the runoff coefficient: the parameter $kr$, ranging from 0 to 1. It drives precipitations towards the groundwater reservoir for values close to 0 and towards the river for values close to 1. It





can therefore be prescribed from the Index of Development and Persistence of the River networks[2], termed IDPR (Mardhel et al., 2004). Indeed, this index compares a theorical network of surface water drainage with the natural hydrological network and thus quantifies the capacity of soils and underlying rocks to encourage rainfall infiltration or diversion to natural stream channels. Since the IDPR ranges from 0 to 2000, with values that decrease as water infiltration increases, we can linearly

index the model parameter $kr$ on the IDPR at the mesh scale. The spatial variability prescribed over the Loire and the Durance catchments is shown in Fig. 4b and 5b.

### 4.2.3   The sub-basin pattern for parameters derived from constraint

Over the Loire and the Durance catchments, we estimated the long-term mean annual streamflow and the long-term mean monthly streamflow for each validation station with a mathematical method. For the long-term mean annual streamflow, we

did a stepwise regression with around 40 physio-climatic catchment descriptors detailed in B. Over the Loire catchment, the long-term mean annual streamflow is then related to the maximum accumulated monthly surplus (Williams et al., 2012) and the maximum daily precipitation. Over the Durance catchment, this is explained by the long-term mean annual precipitation and the long-term mean annual potential evapotranspiration, which are the usual descriptors (Turc, 1954; Mezentsev, 1955; Budyko, 1974; Pike, 1964) besides the relative standard deviation of annual potential evapotranspiration. Afterwards, the

residuals $\frac{\overline{Q_{est}} - \overline{Q_{obs}}}{\overline{Q_{obs}}}$ were submitted to a top-kriging (Skøien et al., 2006), leading to the results shown in Fig. 7. The values mathematically estimated for the Loire's V53 validation stations do not out-perform the initial simulation (see section 4.1). However, those of the Durance's V17 validation stations proved to be more accurate. Therefore, the long-term mean annual streamflow estimations $\overline{Q_{est}}$ of the Durance catchment are used as constraints to calibrate the water balance correction parameter ($Cp$) at the V17 validation stations. In practice, the relative error $\frac{\overline{Q_{sim}} - \overline{Q_{est}}}{\overline{Q_{est}}}$ is added as a fifth objective function

during calibration. Since the water balance correction parameter is still calibrated at the C17 calibration stations over observed streamflows, this model parameter is spatialised at the sub-basin scale of all the stations, as presented in Fig. 5c.

For the long-term mean monthly streamflow, we conducted the method proposed by Sauquet et al. (2008) over each of the two catchments. To do so, (i) we applied a principal component analysis (PCA) over the physio-climatic catchment descriptors, (ii) we carried out a PCA over the Pardé coefficients (Pardé, 1933), which are the 12 non-dimensionnal monthly streamflows

$\frac{\overline{Q_{month}}}{\overline{Q_{annual}}}$, (iii) we established relationships over the calibration sample between these principal components and (iv) we used these relationships to estimate the monthly streamflow at the validation stations. As shown in Fig. 8, this method provides better estimations than the initial regionalisation of the MORDOR-TS model over the Loire catchment. This is not the case for the Durance catchment whose seasonality is already very well constrained by its nival character. As for the Durance's long-term mean annual streamflow, the Loire's monthly streamflow estimated at each validation sample is therefore used as a constraint.

In practice, the two model parameters driving the flow seasonality, *i.e.* the groundwater reservoir draining parameter ($lkn$) and the capacity of the root and the capillarity zones ($ZUmax$) are calibrated at the V53 validation stations with a fifth objective function, the root-mean-square error between the simulated and the estimated long-term mean monthly streamflows. Since the

---

[2]https://www.esrifrance.fr/iso_album/p30_brgm.pdf





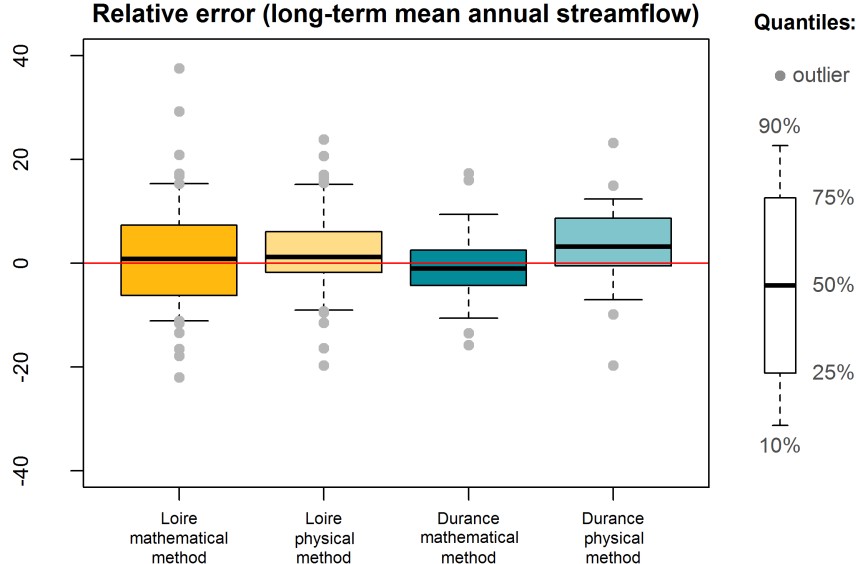

**Figure 7.** Benchmark of the long-term mean annual streamflow in terms of relative error (%). The physical method corresponds to the initial MORDOR-TS simulation with a single sub-basin pattern (cf. 4.1).

two model parameters are still calibrated at the C53 calibration stations over observed streamflows, they are spatialised at the sub-basin scale of all the stations, as presented in Fig. 4c.

### 4.2.4 The physio-climatic sub-basin pattern for parameters derived from transposition

The initial calibration sub-basin pattern (cf. 4.1) can be improved to prevent small validation basins inheriting parameters
from huge calibration basins. To do so, the calibration sub-basin pattern is rearranged with physio-climatic information to become a physio-climatic calibration sub-basin pattern. The validation stations whose drainage area ratio with the downstream calibration station is lower than 20% no longer inherit parameters from this calibration station but inherit parameters from the most similar calibration station in terms of physio-climatic descriptors. The selection of the new donor calibration station is carried out through a Euclidian distance calculated over the principal components of the physio-climatic descriptors. The new
transposition patterns are presented in Fig. 4d and 5d. It is intended for all the remaining parameters about which we have no information or assumptions.

### 5 Results

The single transposition pattern presented in section 4.1 constitutes the initial reference in terms of the distributed parameters. It is referred to as *Exp1*. The tailor-made pattern method developed in this paper consists in the four parameter patterns presented
in section 4.2, namely a uniform pattern, a hydrological mesh pattern, a sub-basin pattern and a physio-climatic calibration sub-





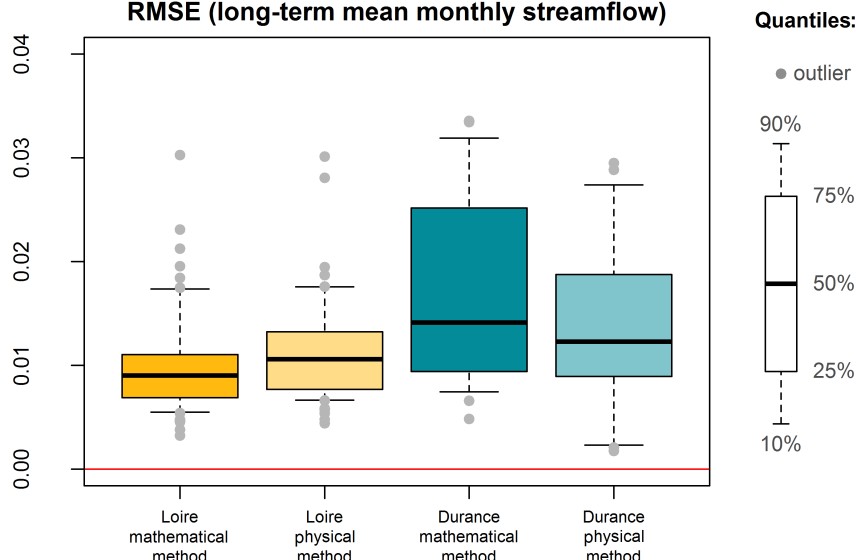

**Figure 8.** Benchmark of the long-term mean monthly streamflow in terms of root-mean-square error. The physical method corresponds to the initial MORDOR-TS simulation with a single sub-basin pattern (cf. 4.1).

basin pattern. This parameter scheme is referred to as *Exp2*. The two experiments are compared based on their performance over the four runoff signatures (daily runoff, seasonality, flood and low flow) in an ungauged context. The performance of the experiments is then analysed over the validation sample with the cumulative distribution functions of KGE. Figure 9 presents these results over the validation sample of the Loire catchment (V53) while Fig. 10 presents those obtained over the validation

sample of the Durance catchment (V17). The closer the distribution function is to the vertical line equal to 1, the better the performance. For better hindsight on the results, a grey area indicates the gap between a uniform set of parameters and a gauged modelling of all the validation stations. For further detail, interested readers may refer to Rouhier et al. (2017).

Over the Loire catchment, the multi-pattern method allows one to significantly decrease the number of model parameters to be calibrated. From 12 parameters for *Exp1*, only five parameters still require calibration for *Exp2*. Despite this drastic

simplification, modelling the four runoff signatures is improved. Although the head of the KGE daily runoff and KGE low flow distributions are slightly degraded, this loss of performance is largely compensated by a significant enhancement of the 50% least well-modelled stations. Seasonality suffers from a slight performance decrease in the middle of its distribution but, as for the two other signatures, the least well-simulated stations are better modelled. For flood, the tailor-made pattern method is even more efficient: regionalising the parameters differently improves the whole KGE flood distribution.

To quantify this performance improvement brought by the multi-pattern regionalisation compared to the single regionalisation, we propose the enhancement index EI defined by Eq. 1.

$$EI = \frac{area\ KGE(Exp1) - area\ KGE(Exp2)}{area\ KGE(Exp1) - area\ KGE(Gauged)} \tag{1}$$




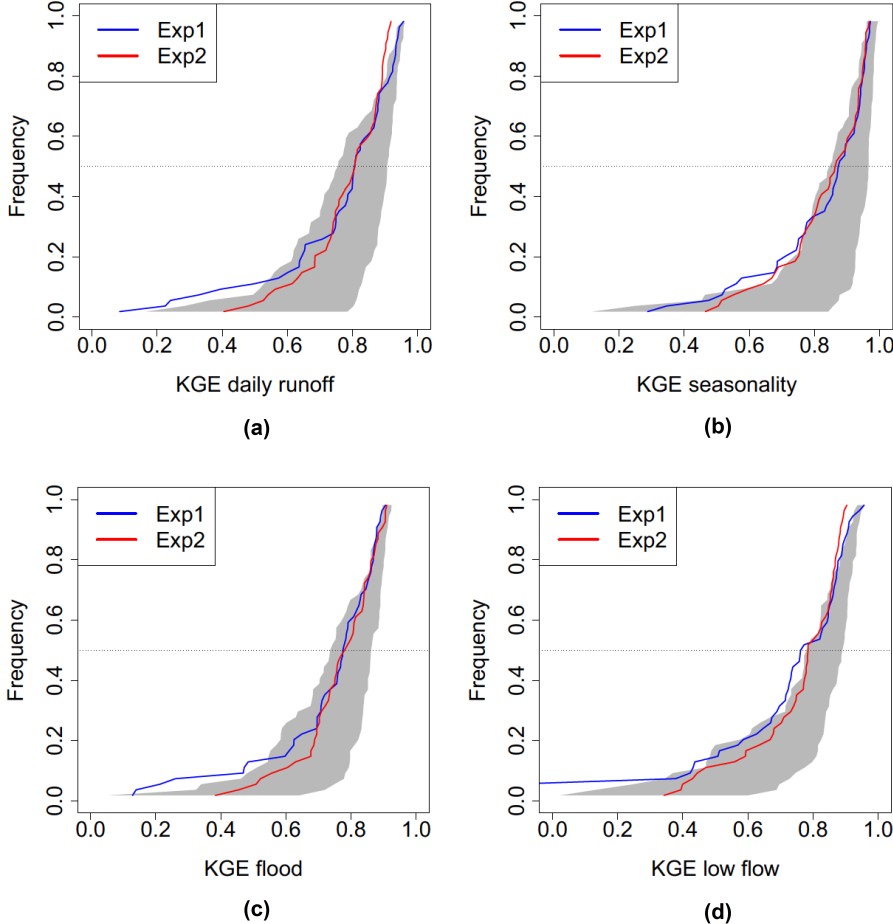

**Figure 9.** Cumulative distribution functions of performance over the validation sample V53 of the Loire catchment to assess the impact on: (a) daily runoff, (b) seasonality, (c) flood and (d) low flow.

The area of an experiment refers to the area under its cumulative distribution function. The greater the improvement towards gauged modelling (right border of the grey area) the closer to 100% it is. The values of the enhancement index over the validation station sample are given in Table 1. Over the Loire catchment, the multi-pattern regionalisation method proves to be very efficient. The improvement towards the gauged modelling is patent and even reaches 41% for low flow.

5    Over the Durance catchment, the number of degrees of freedom is also reduced. From 16 parameters for *Exp1*, only 11 parameters still need to be calibrated for *Exp2*. Over the runoff signatures, the first three are improved. The simulation of daily runoff, seasonality and flood is particularly enhanced for the 50% of the least well-modelled stations. However, these improvements are at the expense of a slight performance degradation for the best modelled stations. Finally, the multi-pattern method is much more debatable for low flow. The degradation of the performance leads to deviation from gauged modelling.



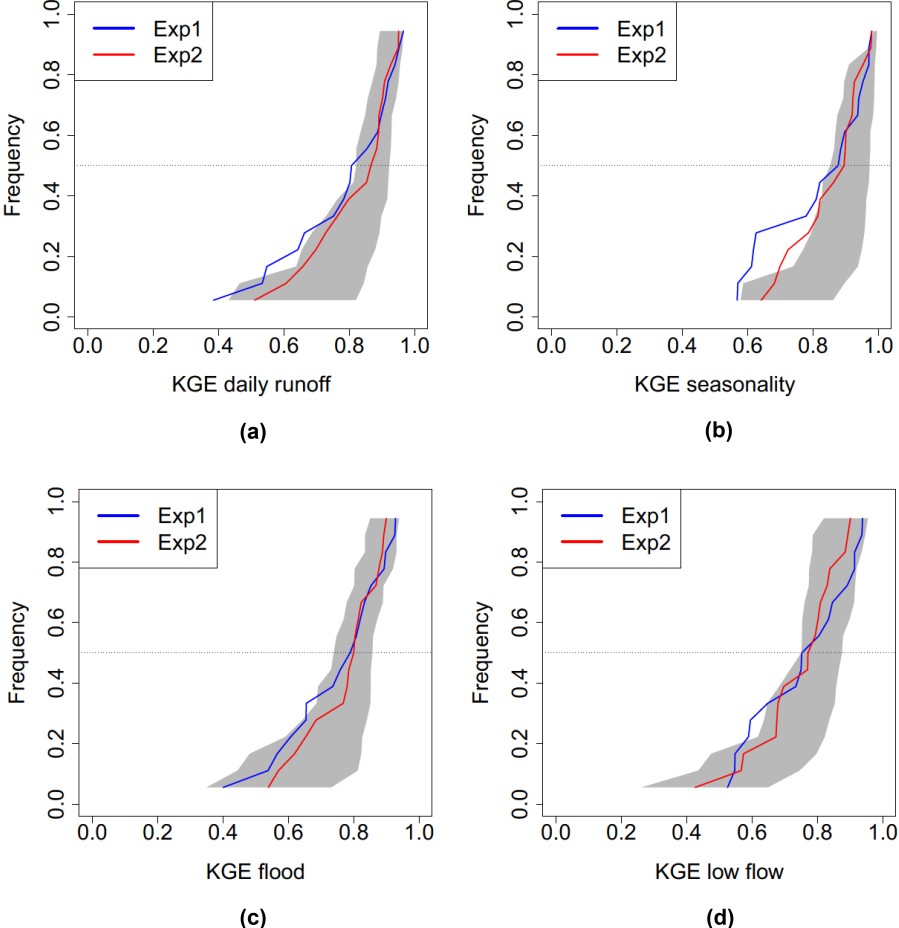

**Figure 10.** Cumulative distribution functions of performance over the validation sample V17 of the Durance catchment to assess the impact on: (a) daily runoff, (b) seasonality, (c) flood and (d) low flow.

Despite this significant loss of performance for low flow, the improvements brought by the tailor-made pattern method are substantial for daily runoff, seasonality and flood with a gain of more than 20% towards gauged modelling.

To obtain hindsight on parameter spatial variability, here we give the enhancement index in relation to a uniform parameter set (left border of the grey area). Over the Loire catchment, the improvement of *Exp2* in terms of daily runoff, seasonality, flood and low flow accounts for 28, 19, 38 and 23%, respectively, of the gap between uniform parameters and gauged modelling. Similarly, the improvement accounts for 32, 16, 44 and 36% over the Durance catchment.





**Table 1.** Summary for the validation station sample of the enhancement index in relation to a single regionalisation method for all parameters.

|              | Loire catchment | Durance catchment |
|--------------|-----------------|-------------------|
| Daily runoff | 18 %            | 25 %              |
| Seasonality  | 8 %             | 23 %              |
| Flood        | 30 %            | 22 %              |
| Low flow     | 41 %            | -10 %             |

## 6 Conclusion and perspectives

This paper aimed to present an unconventional regionalisation scheme where several spatial patterns allowed by several regionalisation methods are adopted depending on the characteristics and the hydrological meaning of the model parameters. Firstly, using a prior sensitivity analysis, the insensitive and equifinal parameters are prescribed uniformly to their default values. Secondly, parameters linked with a physical characteristic are prescribed at the mesh scale. Thirdly, parameters linked with a proxy runoff signature are constrained during the calibration process at the sub-basin scale. Finally, parameters about which we have neither information nor assumptions are transposed according to a physio-climatic pattern constructed over the calibration sub-basins and taking into account their physio-climatic similarity.

This multi-pattern method is evaluated over four runoff signatures of pseudo-ungauged stations over the Loire and the Durance catchments. It not only greatly reduces the number of model parameters to be calibrated, but also proves to be significantly efficient for singular stations. Indeed, modelling of the 50% least well-modelled stations is largely improved for all the runoff signatures, except the Durance's low flow. This outcome suggests that the tailor-made pattern method tends to guarantee a minimum performance in an ungauged context. Whatever the runoff signature and the catchment, the KGE is least of 0.4. The improvement of the tail stations sometimes goes along with the performance degradation of the best modelled stations. However, degradation remains very limited, which does not question the strategy. If we put aside the Durance's low flow, this multi-pattern strategy achieves from 8% to 41% of the distance to gauged modelling compared to a single regionalisation method.

The loss of performance of the Durance's low flow would deserve further research. Understanding the reasons for this degradation could point to a new avenue for improving the regionalisation method. Moreover, the constraint method was restricted to proxy runoff signatures while other types of proxy data such as snow satellite observations could give additional spatialised information. It would therefore be interesting to study whether the integration of these data could bring further improvements to distributed hydrological modelling. Finally, these outcomes stem from the analysis of only two basins. It is therefore necessary to extend the novel method presented here to more catchments.





## Appendix A: Details on the model parameters to be estimated in the study

Table A1 gives details about the model parameters.

## Appendix B: Physio-climatic descriptors

Tables B1 and B2 provide the list of all the physiographic and climatic descriptors used in this paper.

5   *Acknowledgements.*   This work follows from a PhD funded by EDF as part of a CIFRE convention with Sorbonne University during 2015-2018. We would like to thank the BRGM for providing the IDPR over the two catchments studied.



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





**Table A1.** Symbols, descriptions and units of the parameters used in the study.

| Module | Parameter | Description | Unit | Loire | Durance |
|---|---|---|---|---|---|
| Water balance | Cetp | Potential evapotranspiration correction factor | - | x | x |
|  | Cp | Precipitation correction factor | - |  | x |
|  | gtz | Air temperature gradient | °/100m |  | x |
|  | ZUmax | Maximum capacity of the root zone U plus the capillarity storage Z | mm | x | x |
|  | kzu | Distribution between the root zone U and the capillarity storage Z | - | x | x |
| Runoff production | Lmax | Maximum capacity of the hillslope zone L | mm | x | x |
|  | evl | Outflow exponent of storage L | - | x | x |
|  | kr | Runoff coefficient | - | x | x |
|  | evn | Outflow exponent of storage N | - | x | x |
|  | lkn | Logarithm of the outflow coefficient of storage N | $mm.h^{-1}$ | x | x |
| Simplified snow model | kf | Constant part of melting coefficient | $mm.°C^{-1}.day^{-1}$ | x | x |
|  | kfp | Variable part of melting coefficient | $mm.°C^{-1}.day^{-1}$ |  | x |
|  | lts | Smoothing parameter of snow pack temperature | - | x | x |
|  | efp | Additive correction of rain/snow partition temperature | °C |  | x |
|  | eft | Additive correction of melting temperature | °C |  | x |
| Routing scheme | cel | Wave celerity | $m.s^{-1}$ | x | x |
|  | dif | Wave diffusion | $m^2.s^{-1}$ | x | x |





**Table B1.** Physio-graphic descriptors

| Type | Descriptor | Unit |
|---|---|---|
| Physio-graphic | Surface | $km^2$ |
| | Mean elevation | m.a.s.l. |
| | Mean slope | % |
| | Mean hydraulic length | $km$ |
| | Drainage density | $km^{-1}$ |
| | Elongation | - |
| Pedological | Several land uses (between 7 and 8 land uses) | % |
| | Depth to rock | $cm$ |
| | Upper bound of the available water capacity | $mm$ |
| | Lower bound of the available water capacity | $mm$ |
| | Index of development and persistence of the river networks | - |
| | Very permeable surface | % |
| | Medium permeable surface | % |
| | Very impermeable surface | % |

**Table B2.** Climatic descriptors

| Descriptor | Unit |
|---|---|
| Long-term mean annual precipitation | $mm/year$ |
| Long-term mean annual potential evapotranspiration | $mm/year$ |
| Long-term mean annual crop evapotranspiration | $mm/year$ |
| Maximum daily precipitation | $mm/day$ |
| Relative standard deviation of daily precipitation | - |
| Relative standard deviation of monthly precipitation | - |
| Relative standard deviation of annual precipitation | - |
| Relative standard deviation of daily potential evapotranspiration | - |
| Relative standard deviation of monthly potential evapotranspiration | - |
| Relative standard deviation of annual potential evapotranspiration | - |
| Precipitation flashiness | - |
| Potential evapotranspiration flashiness | - |
| Out-of-phase seasonality between precipitation and potential evapotranspiration | - |
| Aridity index | - |
| Maximum accumulated monthly surplus (Williams et al., 2012) | $mm$ |
| Seasonal surplus index (Williams et al., 2012) | $mm$ |