# Peer review of "Tailor-made spatial patterns for hydrological model parameters combining regionalisation methods"

_Hydrology and Earth System Sciences, 2018_

## Short Comment (SC1) · 8 Jul 2018

**Minor comments:**

•        The authors assessed model parametrization and regionalization of a spatially distributed hydrologic model over two mesoscale French catchments. This is a very timely and important topic for spatial hydrology and conventional model calibration using only streamflow data. However, from the manuscript the reader can see only temporal aspects and metrics such as KGE, RMSE and Relative error i.e. all based on discharge (Q) [Page 4 line 6]. Have you noticed the new spatial metric SPAEF (Koch et al., 2018) and other metrics on spatial performance of the models (Rees, 2008)?.

•        The newest paper cited in the introduction section is from the year 2015 by Silvestro et al.

•        The reference list seems not including papers from 2018. I could count only 2 papers from 2015, one paper from 2016, two papers from 2017 (one from Rouhier). However, there has been a lot done in 2018 about spatial parametrization and patterns in hydrology.

•        Page 3, line 17: "The hydrological model used in this work is the spatially distributed conceptual model named MORDOR-TS". The model can be distributed or conceptual, how they can be both?

•        Based on Rouhier et al. (2017), MORDOR-SD is the semi distributed version of the model. However, it is difficult to understand the term "mesh" in that paper. In the section 2.1 of Rouhier et al. (2017), it is clearly mentioned that each mesh corresponds to "elementary sub-catchments". This makes the model again semi-distributed not fully distributed unless the mesh size is unique for every catchment. Please clarify the readers' mind and mention that mesh is not HRU etc.

•        What is mesh size used? Is it triangular? Similar to square grids? The main reason for a semi-distributed model is that a parameter is uniform over the mesh (sub-catchment). However, a fully distributed model like mesoscale Hydrologic Model (mHM, Samaniego et al., (2017)), grids (1x1km etc) are elementary units of simulation and do not vary for different sub-catchments.

•        Rouhier et al. (2017) also uses KGE (a temporal metric for streamflow) and difficult to understand how spatial pattern can be ensured in the regionalization.

•        Similarly, Rouhier et al. (2017) already states in the conclusion that

"The simulated hydrological response was evaluated in light of four runoff signatures in order to finely assess the impact of the spatial variabilities. However, it did not reveal the greater sensitivity of a particular hydrological signature."

In short, looking only runoff may hinder the spatial patterns of the parameters that are sensitive to soil/vegetation related processes such as AET.

•        IDPR appears in figure 5 at page 9 but firstly used in text at page 11.

What is the relation of "Index of Development and Persistence of the River networks" term with parameter regionalization?

•        Page 9 line 3: Sobol for a model with 22 parameters is an extremely heavy work and no details are given except for the names of the sensitive parameters.

•        How many runs were necessary for sensitivity analysis (SA)? How did you conduct the SA? How much time required for the run?

• Which objective functions were tested in the SA?

• Based on the statement at page 16 line 18, low flow performance in Durance basin hampered. How this is related to the way sensitivity analysis done? May be only high flow metrics are used and only high flow related parameters are highlighted as sensitive.

• Selecting appropriate objective function (OF) is crucial in SA (Demirel et al 2018, Koch et al 2018)

• Section 4.2.2 is very interesting. When PET is corrected by a uniform parameter Kc or linked to uniform NDVI, the reader can be curious how a spatially heterogeneous pattern can be formed?

• For that spatially distributed Kc based on LAI patterns should be multiplied by the PET, see Demirel et al., (2018) for the new approach used for PET correction.

• Papers by C. Corbari (Corbari et al., 2015) can be relevant to your work too.

• Other papers are mentioned in the reference list.

**References:**

Cai, G., Vanderborght, J., Langensiepen, M., Schnepf, A., Hüging, H., Vereecken, H., 2018. Root growth, water uptake, and sap flow of winter wheat in response to different soil water conditions. Hydrol. Earth Syst. Sci. 22, 2449–2470.

Corbari, C., Mancini, M., Li, J., Su, Z., 2015. Can satellite land surface temperature data be used similarly to river discharge measurements for distributed hydrological model calibration? Hydrol. Sci. J. 60, 202–217.

Demirel, M.C., Mai, J., Mendiguren, G., Koch, J., Samaniego, L., Stisen, S., 2018. Combining satellite data and appropriate objective functions for improved spatial pattern performance of a distributed hydrologic model. Hydrol. Earth Syst. Sci. 22, 1299–1315.

Koch, J., Demirel, M.C., Stisen, S., 2018. The SPAtial EFficiency metric (SPAEF): multiple-component evaluation of spatial patterns for optimization of hydrological models. Geosci. Model Dev. 11, 1873–1886.

Koch, J., Mendiguren, G., Mariethoz, G., Stisen, S., 2017. Spatial Sensitivity Analysis of Simulated Land Surface Patterns in a Catchment Model Using a Set of Innovative Spatial Performance Metrics. J. Hydrometeorol. 18, 1121–1142.

Kumar, R., Samaniego, L., Attinger, S., 2013. Implications of distributed hydrologic model parameterization on water fluxes at multiple scales and locations. Water Resour. Res. 49, 360–379.

Mendiguren, G., Koch, J., Stisen, S., 2017. Spatial pattern evaluation of a calibrated national hydrological model – a remote-sensing-based diagnostic approach. Hydrol. Earth Syst. Sci. 21, 5987–6005.

Rees, W.G., 2008. Comparing the spatial content of thematic maps. Int. J. Remote Sens. 29, 3833–3844.

Rouhier, L., Garavaglia, F., Le Lay, M., Michon, T., Castaings, W., Le Moine, N., Hendrickx, F., Monteil, C., Ribstein, P., 2018. Tailor-made spatial patterns for hydrological model parameters combining regionalisation methods. Hydrol. Earth Syst. Sci. Discuss. 1–23.

Rouhier, L., Le Lay, M., Garavaglia, F., Le Moine, N., Hendrickx, F., Monteil, C., Ribstein, P., 2017. Impact of mesoscale spatial variability of climatic inputs and parameters on the hydrological response. J. Hydrol. 553, 13–25.

Samaniego, L., Kumar, R., Mai, J., Zink, M., Thober, S., Cuntz, M., Rakovec, O., Schäfer, D., Schrön, M., Brenner, J., Demirel, M.C., Kaluza, M., Langenberg, B., Stisen, S., Attinger, S., 2017. Mesoscale Hydrologic Model. GitHub.

Stisen, S., Koch, J., Sonnenborg, T.O., Refsgaard, J.C., Bircher, S., Ringgaard, R., Jensen, K.H., 2018. Moving beyond runoff calibration - Multi-variable optimization of a surface-subsurface-atmosphere model. Hydrol. Process.

Wambura, F.J., Dietrich, O., Lischeid, G., 2018. Improving a distributed hydrological model using evapotranspiration-related boundary conditions as additional constraints in a data-scarce river basin. Hydrol. Process. 32, 759–775.

Zink, M., Mai, J., Cuntz, M., Samaniego, L., 2018. Conditioning a Hydrologic Model Using Patterns of Remotely Sensed Land Surface Temperature. Water Resour. Res. 54, 2976–2998.

---

## Referee Comment (RC1) · Anonymous Referee #1 · 15 Dec 2018

Review for 'Tailor-made spatial patterns for hydrologic model parameters combining regionalisation methods' by Rouhier et al.

Main Comments

This paper proposes a regionalization approach to identify parameters of a spatially distributed conceptual hydrologic model. The proposed method is compared against standard calibration using four performance measures related to streamflow. Results claim that the proposed regionalization method performs better than the calibration approach by (i) reducing the number of parameters required for calibration, and (ii) by improving performance in most measures. Although the idea of the paper is interesting, in its present form, the paper suffers from several issues which should be addressed before an assessment can be made regarding the reliability of the results. These issues are highlighted below:

1. Writing style: the paper follows a rather confusing writing style. A few issues:
    a. The writing introduces terms such as 'spatialization', 'transposition', 'prescription', etc. for which standard terminology is already available in hydrologic literature. For example, 'prescription' is described as *apriori* parameter estimation (Lines 10, Page 2). It is perhaps better to use the existing term '*apriori* parameter estimation' instead of introducing a new term for the same concept! Similarly, for other terms which may be related to existing terminology, it is better to use that so that readers can easily follow the manuscript. See Blöschl et al. (2013) for terminology related to predictions in ungauged basins and Beven (2001) for standard terminology used in hydrologic modeling.
    b. Grammar and sentence construction can benefit from moderate editing. The paper is a little hard to read right now.
    c. Mixing of results and methods: many results are presented in the methods section. For example Figures 4, 5, 7, and 8 are actually result figures but are presented and discussed in the methods section. A clear separation of methods and results would help organize the manuscript. Another issue is that various sub-plots within Figures 4 and 5 are discussed quite far apart in the manuscript. Generally, figures are presented in order of their appearance in the text. So Fig 4a-d should be discussed prior to Figure 5!
2. Methods: There are differences between how certain methods (such as *apriori* parameter specification) are used in standard literature and how they are interpreted in the study. For example:
    a. Section 4.2.1 implies that 'prescription' is just fixing insensitive parameters to some default value. The introduction, however, states that 'prescription' is *apriori* parameter estimation. The way 'prescription' is implemented in the paper is not *apriori* parameter specification and therefore, either the introduction should be corrected or the method's implementation changed. Section 4.2.2 implements *apriori* parameter estimation consistently with existing literature.
    b. Similarly, the 'constraint' method (Section 4.2.3) is oddly presented. Generally, 'constraining' model parameters implies estimating model parameters as an ungauged site, by using hydro-climatic information (including streamflow) from gauged sites. Constraining typically does not provide a single value for model parameters, rather a range of possible values that reproduce the expected (streamflow) response of the catchment (see Yadav et al. 2007 and Zhang et al. 2008 for more details). However, in the present study, 'constraint' method is applied in an odd manner. It does begin with developing regression relationships between the streamflow predictor (mean annual streamflow or long term mean monthly streamflow) and catchment

charactristics. However, 'in practise' the study just uses observed streamflow data to calibrate relative errors (Lines 19-20, Page 11). Thus, in reality, the study is implementing calibration while labelling it as constraining! To implement constraining, the signature (such as mean annual streamflow) predicted from the regression relationship should be used to reduce the range of (some or all) parameters. Constraining by itself may not identify single parameter sets.

c. Section 4.2.4: It isn't clear why some small validation basins were chosen for application of this fourth approach. Either all validation basins can be parameterized following physio-climatic similarity, or none.

In addition, certain quantifications, such as the Enhancement Index (EI, equation 1) perhaps have established statistical counterparts. For example, the distance between two CDFs is quantified by the Kolmogorov-Smirnov statistic (KS Statistic). This can be used to quantify whether one CDF is significantly different than the other. The statistic not only provides a measure of distance but also whether the distance is significant given the number of data points used to construct the individual CDFs. Given the longer tails of the Exp1 CDFs (blue lines in Figures 9 and 10) it is expected that the area under the curve would be higher. This can be stated as such and introducing a new term to quantify is not really warranted. At the very least, KS-Statistic can be used to complement the EI.

3. Introduction: The introduction states that parameter estimation schemes for distributed models are not as well organized as those for conceptual models. However, there are several papers that have attempted to do this and can be discussed briefly in the introduction. See for example Gotzinger and Bardossy (2007), Samaniego et al. (2010), Wi et al. (2015), and others cited therein.

4. Results: this section is too short, perhaps also because of presentation of results in the methods section. Some more details can be added. For example, why was only Exp1 chosen to compare Exp2? Rouhier et al. (2017) present various methods of calibrating the distributed hydrologic model using various levels of discretization of hydrologic units and climate data. Some of those experiments performed better than Exp1 of the present study (Fig. 8d for KGE low flow in Rouhier et al. 2017 shows some better performances than shown in Figure 9d of present manuscript). Is it possible to use see how the various experiments in Rouhier et al. (2017) compare with the proposed method in this study?

5. Discussion: the paper does not have any discussion section where the results from this study are compared with those from existing works on calibration of distributed hydrologic models. At the very least, some comparison with Rouhier et al. (2017) can be added.

Minor Comments:

1. Abstract: introduces terms 'transposition', 'prescription', etc. without explaining them first. Consider using standard hydrologic modelling terminology for these.
2. Line 22, Page 1: 'comprises' can be 'is represented by'.
3. Lines 2-3, Page 2: provide citation as all three methods (transposition, prescription, and constraint) exist in literature.
4. Lines 28-31, Page 2: discussion of the constraint method is confusing. Multi-objective calibration is not the same as constraining of hydrologic model parameters using runoff signatures (which is mainly used for ungauged basins). Please review this part of the writeup carefully using the information in Yadav et al. (2007) and Zhang et al. (2008).
5. Line 2, Page 3: Which parameter can be calibrated using land surface temperature?

6.  Line 10, Page 3: '50/50 spatial split-sample test' not intuitively clear what this means so a short clarification can help. Or simply the term 'spatial validation' can be used and the more detailed description can follow in the methods section.
7.  Line 31, Page 3: 'nival Durance catchment.' Should be 'nival Durance catchment is used.' Please check for grammar and sentence construction elsewhere.
8.  Line 32, Page 3: 'A' should be 'Appendix A'.
9.  Figures 1 and 2 are the same as those in Rouhier et al. (2017). Please check copyright issues. Please also explain all the terms used in Figure 1, such as P, T, PET, AETbv, etc. in the figure caption.
10. Line 7, Page 4: not clear how long-term mean daily streamflow captures seasonality. It is better to provide equations for each objective function, stating exactly how they are calculated.
11. Please rename section headers for Section 4.1 and 4.2.
12. Section 4.2.1: are sensitivity indices calculated for each calibration station or for the catchment as a whole? Also, is the sensitivity index calculated for all objective functions or just one? If for the entire catchment at once with one objective function, then it is being assumed that sensitivity does not vary across sub-catchments or across objective functions, which may not be true. It is possible that some parameters do not show sensitivity at one site for one objective function but does show sensitivity at another site for another objective function. Please clarify and also provide a table listing sensitivity indices (first-order, interaction effects, and total order) for each parameter so that readers can also assess which parameters are insensitive.

**References**

Beven, K.J., 2001. Rainfall-runoff modeling. The Primer, p.360.

Blöschl, G., Sivapalan, M., Savenije, H., Wagener, T. and Viglione, A. eds., 2013. Runoff prediction in ungauged basins: synthesis across processes, places and scales. Cambridge University Press.

Rouhier, L., Le Lay, M., Garavaglia, F., Le Moine, N., Hendrickx, F., Monteil, C. and Ribstein, P., 2017. Impact of mesoscale spatial variability of climatic inputs and parameters on the hydrological response. Journal of Hydrology, 553, pp.13-25.

Götzinger, J. and Bárdossy, A., 2007. Comparison of four regionalisation methods for a distributed hydrological model. Journal of Hydrology, 333(2-4), pp.374-384.

Samaniego, L., Kumar, R. and Attinger, S., 2010. Multiscale parameter regionalization of a grid-based hydrologic model at the mesoscale. Water Resources Research, 46(5).

Wi, S., Yang, Y.C.E., Steinschneider, S., Khalil, A. and Brown, C.M., 2015. Calibration approaches for distributed hydrologic models in poorly gaged basins: implication for streamflow projections under climate change. Hydrology and Earth System Sciences, 19(2), pp.857-876.

Yadav, M., Wagener, T. and Gupta, H., 2007. Regionalization of constraints on expected watershed response behavior for improved predictions in ungauged basins. *Advances in Water Resources*, *30*(8), pp.1756-1774.

Zhang, Z., Wagener, T., Reed, P. and Bhushan, R., 2008. Reducing uncertainty in predictions in ungauged basins by combining hydrologic indices regionalization and multiobjective optimization. Water Resources Research, 44(12).

---

## Referee Comment (RC2) · Anonymous Referee #2 · 16 Dec 2018

The article proposes a new strategy to regionalize hydrological model parameters. In particular, the new method is a combination of three existing regionalization method. The method is tested using data from two French catchments. The authors claim that the new method shows superior performance and at the same time reduces the number of free parameters. While the results seem to be interesting, I have several major concerns.

1. The authors pick one regionalization method (Exp1) and compare it with the combinedd regionalization method (Exp2). It is possible that Exp1 is not very suitable for the study catchments. The authors should show how the other two regionalization

methods perform in the study catchments.

2. The authors are saying that the new regionalization method is superior to one of the exiting regionalization method. However, I do not think the improvements are really significant. The figures (Figures 9 and 10) show that the new method marginally improves prediction when the baseline performance is low. In fact, for certain cases (Fig. 10d) the performance actually declines.

3. There are countless number of published papers on hydrological modelling. If the authors want to show something new, they need to do more than what they have done. In my opinion, proving something using data from a small/climatologically homogeneous region does not make a lot of sense given that we already know a lot from previous modelling exercises. If the authors want to prove that their conclusions are meaningful, they need to consider a much larger number of catchments situated across climatic and geographic regions.

4. How useful are the conclusions from this study for other hydrological models? Are they also applicable to other models? If not, I am not sure how useful the results from this study are. If yes, can the authors explain why?.

5. The authors are claiming that the new method has helped them in reducing the number of free parameters from 12 to 5. However, 5 is not a small number. Many researchers have argued that a hydrological model with just 4 free parameters can perform well (e.g., Hornberger et al., 1985). Thus, to me, the model is unnecessarily complex even after the simplification exercise.

6. The methods are quite confusing in this study. The authors are saying the new regionalization method is helping in model calibration. Regionalization methods re used for prediction in ungauged/pseudo-ungauged basins. Once the model parameters are calibrated using data from gauged basins, the model should work for the ungauged basins without calibration. The authors need to explain their methods clearly.
Overall, the authors need to substantially expand their analysis to show that their results are meaningful. Presentation can be improved. Often unconventional terms such as specialization are used. The authors need to define them properly before using them.

Reference: Hornberger, G. M., Beven, K. J., Cosby, B. J., & Sappington, D. E. (1985). Shenandoah watershed study: Calibration of a topography‐based, variable contributing area hydrological model to a small forested catchment. Water Resources Research, 21(12), 1841-1850.

---

## Author Comment (AC1) · 14 Feb 2019

**Detailed response to the comments of M. Demirel**

We want to thank M. Demirel for his valuable comments and suggestions about our paper. In this author comment, we give our answers to its remarks.

**[1]** We agree that spatial metrics, such SPAEF proposed by M. Demirel, may be very usefull to calibrate distributed models on spatial observations. SPAEF was developed to compare two maps (for example, actual evapotranspiration over the basin) but here, we are evaluating the streamflow at some pseudo-ungauged locations but not maps. Therefore, we apply the KGE over the whole time series of these pseudo-ungauged stations to assess the performance of the regionalization.

**[4]** The model is conceptual since its parameters do not stand for physical properties of the catchment. They are parameters controlling the several processes of the hydrological cycle described in a conceptual way. On the other hand, the catchment is spatially distributed as it is divided into hydrological meshes (= small subcatchments). Every hydrological mesh is associated with its own set of conceptual parameters and its own climatic inputs and its own state variables.

**[5]** We are aware that the terminology of "semi-distributed" and "distributed" varies from one hydrologist to another. In this paper, the term "semi-distributed" means that climatic inputs are distributed but not the hydrology (the parameters are identical over the whole catchment, and the spatialization is not done according to hydrological sub-catchments : we cannot simulate the streamflow at interior points). On the contrary, the term "distributed" means that both the climate and the hydrology are distributed. The catchment is divided into elementary sub-catchments according to a target size (of 100 km²) and a Digital Elevation Model.  In the distributed model, a streamflow simulation can be obtained at any mesh oulet of the model. Then, even if we do not use regular meshes (square or triangular grids) but irregular meshes derived from the DEM, the  model may be qualified as spatially distributed.

**[6]** The target size of the meshes is 100 km². To divide the catchment into hydrological meshed, an algorithm then tries to minimize the variance of the area of the meshes around the target size. Therefore, the hydrological meshes are sub-catchments of approximately 100 km².

**[7]** To assess the spatial performance, we apply the KGE over validation stations whose streamflow is never used to calibrate the parameters. We assess the performance at pseudo-ungauged stations over their whole streamflow time series. Therefore, since theirs streamflow time series are never used to calibrate the parameters, theirs KGE quantify the performance of the parameter regionalization.

**[8]** We agree with M. Demirel that use of other observations, as AET estimations, may help to better identify some of the parameters.  However, our experience is that on the considered regions, such observations suffer from significant biases. Therefore, including them in the calibration process may degrade parameter estimation and model performances. It is why we concentrate on streamflow data. Moreover, the four signatures are about runoff but they are related to different hydrological processes. They allow to evaluate the model both in seasonality, current flows, low flows and high flows.

**[9]** The IDPR appears in figure 5 to facilitate the comparison with the other spatial patterns. This index is used to prescribe one of the parameters (as explained in section 4.2.2). So, the IDPR is the basis of the regionalization of one parameters.

**[10-11-12-13-14]** Details about the sensitivity analysis are not are not given in this paper to avoid overloading the article since the sensitivity analysis would deserve a whole paper. However, we can

say that the sensitivity analysis is conducted with a quasi-Monte Carlo method over the four KGE criteria (daily runoff, seasonality, flood and low flow) plus the KGE over the fractional snow cover. A total of 45.056 combinations were tested over each basin and required days of computation.

**[15]** Before (Exp1), the crop coefficient was uniform. In the paper, we propose to make it spatially variable (Exp2), based on the NDVI map obtained by satellite. Kc is therefore spatially heterogeneous. "*Previously*, it was obtained through a one-parameter formulation where the parameter was set uniformly to its default value. *From now on*, the inter-annual time series of NDVI (16-day and 1-km² resolution) is aggregated at the mesh scale and then used to prescribe Kc at the same scale."

**[2-3-16-17-18]** Thank you for the papers suggested, which are really relevant. We added some of them in our paper.

---

## Author Comment (AC2) · 14 Feb 2019

**Detailed response to the comments of referee 1**

We want to thank referee 1 for his accurate and helpful review of our manuscript. In this author comment, we list how each of the remarks provided by the referee was addressed.

1. **Writing style**

a. We agree with you that terminology issues are important. In this paper, we tried to use short terms whose definitions are given at the beginning of the paper. On top of that, we tried as much as possible to use standard terminology, inspired by *Bloschl et al. (2013)*. However, our paper presents a new approach which is sometimes out of the scope of the existent literature. Regarding the term 'prescription', we do not define it as '*apriori* parameter estimation' which, in the context of distributed modelling, usually refers to giving spatialized parameter values based on spatialized catchment characteristics values. Here, 'prescription' is a broader term which encompasses '*apriori* parameter estimation' but also the assignment of a value based on the user experience.
➔ We propose to modify the manuscript as follows (Introduction):
p2, l.10 "Prescription consists in assigning parameter values based on the literature, the user experience on the hydrological model or empirical relationships between model parameters and catchment characteristics"

b. Before submission, the paper was corrected by a native English person.

c. We agree that the structure of our article is a little bit different from conventional papers since the methodology of combining several regionalization methods necessarily implies to describe parameter spatial patterns which can be seen as a kind of results. Therefore, we cannot consider a method/results structure. Instead, we propose a 'parameter spatial patterns' section followed by a 'performance analysis' section. Regarding the order of the figures, we first tried to respect the order of appearance in the text (see "Figure_order.pdf") but we found it very difficult to compare the different parameter spatial patterns. Thus, we chose to make one figure for the Loire basin (Fig. 4) and one figure for the Durance basin (Fig. 5). However, if you believe that it is better to present the figures in their order of appearance in the text as in "Figure_order.pdf", we are quite open to do it.

2. **Methods**

a. We used an awkward wording. Section 4.2.1. consists in fixing insensitive parameters to some values based on the user experience about the hydrological model. On top of that, we modified the introduction as discussed above not to suggest that prescription is only '*apriori* parameter estimation'.
➔ We propose to modify the manuscript as follows (section 4.2.1):
p9, l.8 "Consequently, we propose to prescribe the insensitive parameters uniformly at a value derived from literature or user experience, namely five parameters for the Loire catchment and two for the Durance catchment".

b. Constraint is a little applied method first introduced by Yadav et al. 2007 which can be considered in different ways. We can understand this method as providing a range of possible values. However, we can also understand it as a method based on estimations at ungauged sites, and this is how we considered the constraint method. Indeed, in our view, the constraint method uses streamflow estimations at ungauged sites to determine the parameter sets while the calibration method uses streamflow observations. We thus believe that name it calibration would be quite misleading. That is why we believe it is better to name it 'constraint'. But once again, if you think that this is a mistake, we are open to name it "ungauged calibration".

c. The fourth approach cannot be applied to all validation basins due to the inter-dependance of the basins. For example, consider a validation basin A upstream a calibration basin B, and on another side a validation basin C upstream a calibration basin D. If the most similar basin of A according to physio-climatic characteristics is basin D and the most similar basin of C is basin B, the parameter set of A should be the one calibrated on D and the parameter set of C should be the one calibrated on B.

A-v  [D]-> B-c [B]    ;   C-v –[B] > D-c [D]     parameter set into brackets

However, the calibration of B depends on the parameter set of A (upstream) that is to say the parameter set of D, which depends on the parameter set of C (upstream) that is to say the parameter set of B. We should therefore do several calibration passes until the parameter sets stabilize. Thus, at the scale of several dozens of catchments, this is not an option given the computation time.

However, as we can see on the figures below, there is a clear pattern that small catchments inheriting parameters from huge calibrations basins have poor performance. In the figures below, the surface ration RS is defined as :

$$RS_i = \frac{Drainage\ area_{validation\ station\ i}}{Drainage\ area_{calibration\ station\ downstream}}$$

[Figure]

Figure 1: KGEq performance according to surface ratio
- Loire@Gien

[Figure]

Figure 2: KGEq performance according to surface ratio
- Durance@Cadarache

According to these figures, we therefore adopted the arbitrary value of RS = 20 %: the basins with a RS <20 % were submitted to the physio-climatic transposition whereas the others were still submitted to the previous upstream-downstream transposition.

➔ We propose to modify the manuscript as follows (section 4.2.4):

"Indeed, according to preliminary attempts of regionalization not detailed here, we found out poor performance for validation stations whose drainage area ratio with the downstream calibration station is low. To address this, we propose to rearrange the calibration sub-basin pattern with physio-climatic information to become a physio-climatic calibration sub-basin pattern. To do so, the idea is to no longer inherit parameters from this calibration station but inherit parameters from the most similar calibration station in terms of physio-climatic descriptors. The selection of the new donor calibration station is carried out through a Euclidian distance calculated over the principal components of the physio-climatic descriptors. As basins are nested ones, applying this method for all the validation basins is not an option since it would implies the inter-dependance of all the parameter sets. Therefore, we adopt the 20% arbitrary value of drainage area ratio meaning that all the validation basins with a drainage area ratio lower than 20% are concerned by this physio-climatic

transposition while the parameters of the basins with a drainage area ratio higher than 20\% continue to be transposed as before, i.e. as described in section 4.1."

Following your suggestion, we applied the KS-Statistic as an alternative to our Enhancement Index. However, results show that it is not well suited to our problem. If we take the example of KGE seasonality over the Durance catchment, we have the figure below. The KS distance between the CDF of Exp1 and the gauged modelling (right border of the grey area) is 0,59 while the distance between Exp2 and the gauged modelling (right border of the grey area) is 0,71. Therefore, this test suggests that Exp1 provides better results than Exp2, wheareas the figure clearly shows the contrary. This is because the KS statistic calculates the vertical maximum absolute difference between the two cumulative distribution functions. Figure 4 shows that the KS test between the CDF of Exp1 and the gauged modelling in blue and the one between the CDF of Exp2 and the gauged modelling in red. So the KS test which does not represent the distance between the whole distribution functions and is not well suited to our analysis.

[Figure]

Figure 3: Cumulative distribution functions of performance for the runoff seansonality over the Durance catchment

[Figure]

Figure 4: Cumulative distribution functions of performance for the runoff seansonality over the Durance catchment with the KS test

**3. Introduction**

Thanks for your suggestions. We added the reference of Gotzinger and Bardossy (2007), which was effectively missing.

**4. Results**

The results section was renamed "Performance analysis" to be more explicit. We also renamed the title of section 4 ""Construction of parameter spatial patterns".
Exp1 corresponds to Exp6 of Rouhier et al. (2017). It is the best solution of Rouhier et al. (2017). Obviously, it is not the best for all signatures and the whole distribution. For example, as you mentioned, this experiment is not the best for low flow (see Rouhier et al. (2017)). However, it is clearly the best compromise of Rouhier et al. (2017) with the best overall performance.
➔ We propose to modify the manuscript as follows (section 5):
" It is referred to as *Exp1* and corresponds to the regionalisation method with the best overall performance of those discussed in Rouhier et al. (2017)."

Regarding the comparison with the various experiments in Rouhier et al. (2017), we do not want to add to many experiments in the present paper as it would make reading difficult. However, you can find on the 4 figures below the comparison with two experiments of the previous paper : LRef (uniform climatic inputs + uniform parameters) and Exp4 (spatialized climatic inputs + uniform parameters), for the Loire basin.
Be careful, the grey area of the present article is different from the one of the previous article.

[Figure]

[Figure]

[Figure]

**5. Discussion**

As we explained before, we believe that it is not a good idea to add the experiments of Rouhier et al. (2017). On top of that, the purpose of the present article is not to compare climatic inputs spatialization and parameter sets spatialization. Nevertheless, we could add a supplementary material with the 4 figures above and three enhancement indexes:

- Enhancement index about climatic inputs spatialization:

$$EI_c = \frac{area\ KGE(LRef) - area\ KGE(Exp4)}{area\ KGE(LRef) - area\ KGE\ (gauged)}$$

where the gauged experiment corresponds to the right border of the grey area.

- Enhancement index about parameter sets spatialization:

$$EI_p = \frac{area\ KGE(Exp4) - area\ KGE(Exp2)}{area\ KGE(LRef) - area\ KGE\ (gauged)}$$

|  |  | EIc | EIp |
|---|---|---|---|
| **Loire** | Daily runoff | 64% | 10% |
|  | Seasonality | 62% | 7% |
|  | Flood | 61% | 15% |
|  | Low flow | 73% | 6% |
| **Durance** | Daily runoff | 67% | 11% |
|  | Seasonality | 69% | 5% |
|  | Flood | 64% | 16% |
|  | Low flow | 69% | 11% |

According to these results, climatic inputs spatialization provides an enhancement of about 65% while parameter sets spatialization brings an enhancement of about 10%, in ungauged context.

Let us know if you consider that this supplementary material may be relevant.

➔ We propose to modify the manuscript as follows (section 6):

"As for Gotzinger and Bardossy (2007), our paper thus shows that combining several regionalisation methods make it possible to benefit from the advantages of each to provide better performances in ungauged basins."

6. **Minor comments**

   1. See 1.a.

   2. Changed for 'represented by'

   3. Quite a number of citations are given in the paragraphs below (page 2 - line 5 to page 3 - line 14)

   4. See 2.b.

   5. Silvestro et al. (2015) don't calibrate one parameter with land surface temperature but the parameter sets.

   6. Changed to 'spatial validation' .

   7. There is already a verb in the sentence: 'comprises'.

   8. Corrected

   9. Added

   10. It is not possible to provide equations for each objective function since for some of them it is not only an equation but an algorithm. However, we can illustrate this in a supplementary material with the figures below :
   >> Daily Runoff

[Figure]

**Figure 5: Daily runoff**

>> Seasonality
The hydrological signature on which the criterion is a time series of 365 values, the first value being the average of the discharges of every January 1st.

[Figure]

**Figure 6: long-term mean daily streamflow**

>> Flow duration curve
KGE is applied to every monthly flow duration curve. Then, the 12 KGE values are aggregated with a weighted mean for which the weights depend of the monthly means, as

$$KGE_{flood} = \sum_{i=1}^{12} \frac{\overline{Q_i}}{\sum_{j=1}^{12} \overline{Q_J}} KGE_i$$

[Figure]

**Figure 7: Flow duration curve of January**

[Figure]

**Figure 8: Flow duration curve of August**

>> Flow recessions

The selection of the recessions of the daily streamflow time series result of: (i) the smoothing of the time series with a 7-day window, (ii) the selection of the decreasing periods, (iii) the removal of the first 7 days of each decreasing period.

[Figure]

**Figure 9: flow recessions are identified in grey**

11. We did not find satisfying alternatives for these titles. Hence, we propose to keep the titles of sections 4.1 and 4.2 which explain our approach.

12. The sensitivity analysis was done for several stations and for all the four objective functions. Thus, it gave the sensitivity for all the parameters at a given station for a given objective function. It is too ambitious to describe in details this sensitivity analysis here and to provide sensitivity indices. However, a paper might be written about this sensitivity analysis that we cannot sum up here.

➔ We propose to modify the manuscript as follows (section 4.2.1):
"A sensitivity analysis of the MORDOR-TS model has been conducted over hundreds of French catchments according to the approaches for each of the four objective functions according to the approaches of Sobol (1993), Homma and Saltelli (1996) and Liu and Owen (2006)."

Nevertheless, we give some details below :

[Figure]

**Figure 10: Fanova graphs - KGE daily runoff for Loire@Gien**

This figure shows the results obtained for the discharge station of Gien in our 12-parameter configuration as regards KGE daily runoff. This graphical representation is inspired by the FANOVA graphs of Muehlenstaedt et al. (2012). The radius of the red disc reprensents the value of the first order sensitivity Si of the parameter (Sobol, 1993), while the radius of the blue disc gives the total sensitivity STi of the parameter (Homma and Saltelli, 1996), i.e. first-order sensitivity plus its sensitivity in interaction with the other parameters. The red disc is superimposed on the blue disc since the first-order sensitivity is always lower than the total sensitivity. The larger the red disc, the greater the first-order sensitivity. The larger the differences between the blue and the red discs, the greater the interactions with the other parameters. The distribution of these interactions is represented by the blue lines between the parameters. The greater the thickness of the line, the greater the interaction TIIij between the two parameters (Liu and Owen, 2006). Figure 10 therefore informs us that five parameters are not sensitive at all: the snow parameters (kf and lts), the parameter generating the delayed flows (evl), the diffusivity (Dif) and the celerity (Cel). This outcome is confirmed by the sensitivities distributions over the 106 discharge stations of the Loire catchment

shown in Figure 11. The same outcome is obtained for the other three signatures : KGE daily regime, KGE flood and KGE low flow.

[Figure]

**Figure 11: Boxplot of parameter sensitivity over the 106 streamflow stations of the Loire catchment as regards the KGE daily runoff**

**References**

Blöschl, G., Sivapalan, M., Wagener, T., Viglione, A., and Savenije, H.: Runoff Prediction in Ungauged Basins. Synthesis across Processes,Places and Scales, Cambridge University Press, Cambridge, 2013

Homma, T., Saltelli, A., 1996. Importance measures in global sensitivity analysis of nonlinear models. Reliability Engineering & System Safety 52 (1), 1–17.

Liu, R., Owen, A. B., 2006. Estimating mean dimensionality of analysis of variance decompositions. Journal of the American Statistical Association 101 (474), 712–721.

Muehlenstaedt, T., Roustant, O., Carraro, L., Kuhnt, S., 2012. Data-driven kriging models based on fanova-decomposition. Statistics and Computing 22 (3), 723–738.

Sobol, I. M., 1993. Sensitivity estimates for nonlinear mathematical models. Mathematical modelling and computational experiments 1 (4), 407–414.

---

## Author Comment (AC3) · 14 Feb 2019

**Detailed response to the comments of referee 2**

We want to thank referee 2 for his review of our manuscript. In this author comment, we list how each of the remarks provided by the referee was addressed.

1. There are multiple possibilities to regionalize a distributed hydrological model and the present paper does not pretend to be exhaustive at all. On top of that, applying a single prescription method requires having some catchment characteristics for each model parameter which is absolutely not the case. Indeed, for the prescription method, each parameter of each hydrological mesh is prescribed according to the values of some catchment characteristics. However, MORDOR-TS is not a physically based model but a conceptual one. Regarding the only constraint method, it would require many proxy data which should be the subject of a whole article.

   ➔ We propose to add this sentence in the manuscript (Section 5):
   "The present paper does not claim to be exhaustive about the regionalisation methods. In fact, there are multiple possibilities to regionalise a distributed hydrological model with transposition, prescription and constraint. Here, we propose to compare a single transposition with a combination of the three regionalisation methods."

2. We agree that the improvements are not huge. However, prediction in ungauged basins is a very difficult issue which progresses very slowly. The figure below comes from Blöschl et al. (2013) and presents a synthesis of the literature about several regionalization methods for the estimation of annual runoff. We added to this figure two lines (the blue one for Exp1 and the red one for Exp2). According to this figure, our results are in the top part of the regionalization methods and the coefficient of determination rises from 0.92 for Exp1 to 0.94 for Exp2. So, in the context of prediction in ungauged basins, we think that the improvements we obtained with the combination of several regionalization methods are not marginal but significant.

[Figure]

Regarding the performance of Fig 10d, the performance actually declines between Exp1 and Exp2. However, hydrological modelling is always a compromise between several criteria. Here, we tried to have a convenient model for both daily runoff, seasonality, flood and low flow modelling. Nevertheless, if the aim was only to estimate low flows, the same analysis could have been conducted with a single objective function over low flows.

3.  On this point, we do not agree with your remark. In our study, one basin (namely Loire or Durance) comprises hundreds of hydrological meshes with dozens of stations. In fact, the regionalisation of the Loire basin analyses 106 catchments while the regionalisation of the Durance basin analyses 34 catchments. Therefore, the number of catchments used in this paper is significant. On top of that, the regions are not climatologically homogeneous. With its 35 707 km², the Loire basin allows us to cover catchments from pluvio-nival to pluvial ; whereas the Durance basin with its 11 738 km² allows us to cover catchments from nival to nivo-pluvial. Therefore, the area studied within the scope of this paper is neither small nor climatologically homogeneous.

[Figure]

The philosophy of regionalisation developed in this paper aims to be generic and it could be applied to any hydrological models. In this study, this strategy is effectively tested for the only MORDOR-TS model. However, we tried to come out with general recommendations as described by Fig. 6. If the strategy described by this figure can be applied to any hydrological model, it necessarily requires a good understanding and a good knowledge of the model. But we do believe that a relevant regionalisation method can only be achieved if the model is well known, both physically and

numerically. However, testing the methodology over another hydrological model is out of the scope of this paper.

We agree with the idea of parcimony in hydrological modelling. However, the MORDOR-TS model has many parameters because it is intended to be applied to many climatic contexts. Indeed, it is used in the same time for nival basins and for pluvial basins, for glaciarized and for karstic basins, at hourly and daily timesteps. Some parameters were even introduced to compensate bias in precipitation or temperature estimations, to ensure good performances in every operational contexts. On top of that, Garavaglia et al. (2017) showed that having less parameters involves a significant loss of performance when tested on many basins.

4. We do not understand this comment. Can you reformulate it?

5. Concerning the terminology issue, also pointed out by referee #1, we tried to use short terms whose definitions are given at the beginning of the paper. On top of that, we tried as much as possible to use standard terminology, inspired by *Bloschl et al. (2013)*. However, our paper presents a new approach which is sometimes out of the scope of the existent literature. We did not find very satisfying alternatives, but we are open to any relevant suggestion.

References

Blöschl, G., Sivapalan, M., Wagener, T., Viglione, A., and Savenije, H.: Runoff Prediction in Ungauged Basins. Synthesis across Processes,Places and Scales, Cambridge University Press, Cambridge, 2013

Garavaglia, F., Le Lay, M., Gottardi, F., Garçon, R., Gailhard, J., Paquet, E., and Mathevet, T.: Impact of model structure on flow simulation and hydrological realism: from lumped to semi-distributed approach, Hydrology and Earth System Sciences, https://doi.org/10.5194/hess-2017-82, 2017.

---

## Author Comment (AC4) · 14 Feb 2019

**Tailor-made spatial patterns for parameters through regionalization methods combination: improvement of predictions in ungauged basins**

Rouhier Lauraa,b,\*, Le Lay Matthieua, Garavaglia Federicoa, Le Moine Nicolasb, Hendrickx Frédéricc, Monteil Célinec, Ribstein Pierreb

aÉlectricité de France, DTG, Grenoble, France bSorbonne Université, METIS, 75005 Paris, France cÉlectricité de France, R&D, Paris, France

**Abstract**

Calibration of spatially distributed models is a big issue given their overparameterization. Three usual regionalization method can be distinguished which are transposition, prescription and constraint. This paper proposes a strategy where the three methods are combined to provide several spatial patterns according to the model parameters. On the one hand, insensitive and equifinal parameters are prescribed uniformly while "physical" parameters are prescribed at the mesh scale. On the other hand, parameters linked with a proxy runoff signature are constrained over each sub-basin and the remaining parameters are transposed with a physio-climatic pattern constructed over the calibration sub-basins.

The above tailor-made pattern regionalization is applied at the daily time step over two large French catchments, the Loire catchment at Gien covering  $35,707 \text{ km}^2$  and the Durance catchment at Cadarache covering 11,738

Preprint submitted to Elsevier

\*Corresponding author

Email address: laura.rouhier@edf.fr (Rouhier Laura)

km2. It is then evaluated and compared to a single regionalization method over dozens of validation stations, treated as ungauged during the parameter regionalization. For that purpose, simulated and observed streamflows are compared in light of four runoff signatures: daily runoff, daily regime, flood and low flow. The results show that the tailor-made patterns succeed to enhance significantly almost all the signatures. The enhancement appears for the least well-modelled stations which tends to guarantee a minimum performance in the ungauged context.

*Keywords:* parameter spatial variability, distributed hydrological modelling, regionalization, ungauged basins

**1 1. Introduction**

Spatially distributed hydrological models allow for (i) spatially distributed climatic inputs, (ii) spatially distributed model parameters, (iii) ungauged 3 simulations and (iv) upstream-downstream consistency. With the increasing availability of spatial data and the improvements in computational power, 5 this type of model represents a real potential for hydrological modelling. 6 The Distributed Model Intercomparison Project (Smith et al., 2004; Reed 7 et al., 2004; Smith et al., 2012, 2013) investigated the capabilities of exist-8 ing distributed hydrologic models. However, this project did not provide any 9 recommandation about parameter estimation schemes. The strategy is not as 10 well defined as for lumped models whose parameters usually follow from cal-11 ibration over the observed outlet streamflow. Indeed, in distributed models, 12 each spatial unit comprises one set of parameters while most of these units 13 are ungauged (Sivapalan et al., 2003; Hrachowitz et al., 2013). Distributed 14

15 models therefore suffer from overparameterization and equifinality (Beven
16 and Hornberger, 1982; Beven, 2001). To overcome these difficulties, one can
17 rests upon three regionalization methods: (i) transposition, (ii) prescription
18 and (iii) constraint.

19 Transposition consists in grouping the  $N_u$  spatial units into  $N_r$  regions, 20 each of them comprising one set of  $N_p$  parameters calibrated over gauged 21 discharge stations. The region delineation can follows from physio-climatic 22 similarity (Beldring et al., 2003; Kumar et al., 2013) or gauged network (An-23 dersen et al., 2001; Feyen et al., 2008; Khakbaz et al., 2012; De Lavenne 24 et al., 2016). This method reduces the dimensionality of the optimization 25 problem from  $N_u \times N_p$  to  $N_r \times N_p$  free parameters.

Prescription is based on *a priori* or empirical relationships between catch-26 ment characteristics and model parameters (Koren et al., 2000; Twarakavi 27 et al., 2009). That way, Andersen et al. (2001) and Khakbaz et al. (2012) 28 tested an uncalibrated model with distributed parameters directly estimated 29 from field data, literature and previous studies. However, within the frame-30 work of distributed modelling, prescription is pretty often enhanced with 31 transposition to reduce the gap between the modelling and the physical ex-32 pertise (Francés et al., 2007; Smith et al., 2013). Francés et al. (2007); Pokhrel 33 and Gupta (2010); Samaniego et al. (2010) and Khakbaz et al. (2012) first 34 prescribed spatial parameter fields from catchment characteristics and then 35 adjusted them through transposition of uniform correction coefficients (*i.e.* (i.e.36 calibration of one region) called superparameters or global parameters cali-37 brated over the observed outlet streamflow. For instance, Pokhrel and Gupta 38 (2010) defined three superparameters per model parameter: a multiplying, 39

an additive and a power coefficients involving the calibration of 3  $\times N_p$  su-40 perparameters. The two steps can also be inverted by first calibrating the 41 model parameters uniformly and then modifying them with spatial fields es-42 timated from catchment characteristics without further calibration (Koren 43 et al., 2004; Khakbaz et al., 2012). To a more limited extent, prescription 44 can also be a tool to calibrate the model parameters to be transposed. As 45 proposed by Ajami et al. (2004), each region of the catchment can be cal-46 ibrated over its observed outlet streamflow by temporarily prescribing the 47 downstream parameters with catchment characteristics. 48

Finally, constraint relies on proxy data, *i.e.* on hydrological signature 49 estimated without any streamflow measure that can give a clue about the 50 catchment hydrological behaviour. Constraint consists in using these proxy 51 data instead of streamflow time series as a constraint in the calibration pro-52 cess. Madsen (2003) appraised a multi-objective calibration of a distributed 53 model over observed outlet streamflow and groundwater levels measured at 54 17 interior wells. Instead of groundwater data, Khan et al. (2011) and Sil-55 vestro et al. (2015) proposed to calibrate the distributed model parameters 56 with remote-sensing data. Along with streamflow observations, Khan et al. 57 (2011) used satellite-derived flood maps to calibrate a module of a distributed 58 model designed for flood, while Silvestro et al. (2015) proved the usefulness 59 of land-surface temperature and surface soil moisture satellite observations 60 to reduce parameter equifinality. 61

This paper aims to advance one step further and proposes to combine the three regionalization methods. The model parameters are spatialised with one of the three methods according to their characteristics and hydrological 65 meaning. It follows from this multi-method, four parameter spatial patterns: 66 a uniform, a hydrological mesh and two intermediate patterns. The method 67 is applied over two French mesoscale catchments, the Loire at Gien and the 68 Durance at Cadarache, for the 1980-2012 period. Thanks to a 50/50 spatial 69 split-sample test, the performance of the tailor-made patterns is assessed over 70 pseudo-ungauged stations and compared with that of a unique transposition 71 scheme.

The paper is organised as follows. Section 2 presents the distributed rainfall-runoff model and the evaluation criteria. Section 3 introduces the data set. Section 4 details the parameter spatial patterns, Section 5 discusses the results and section 6 provides conclusions and perspectives.

**76 2. Modelling**

**77 2.1. Distributed rainfall-runoff model**

The spatially distributed rainfall-runoff model used for this study is the conceptual MORDOR-TS model presented in Rouhier et al. (2017). The catchment is divided into hydrological meshes, each of them attributed to one set of parameters and connected to each other according to the hydrographic network. At each daily time step, the continuous model (i) calculates the water production of each mesh independently and (ii) routes all production to the simulation points, which can be any mesh outlet.

The production module aims at quantifying the exchanges between different components of the hydrologic cycle. Based on precipitation and air temperature data, six conceptual interconnected storage components evolve and supply the hydrographic network as described by Figure 1a. The ver-

(a) Production structure

tical spatialisation of the hydrological meshes into elevation zones, designed for mountainous regions, is only activated for the Durance catchment. A complete description of the production module can be found in Garavaglia et al. (2017).

many runoff contributions are estimated. They are propagated to the simulation points through the hydrographic network as described in Figure 1b.
The routing module combines the intra-mesh and inter-mesh transfers by
means of a formulation based on the 1D diffusive wave model, with celerity and diffusion independent of runoff (Hayami, 1951; Litrico and Georges,
1999).

In its entire formulation, MORDOR-TS has 22 free parameters. In this study, a simplified version is adopted with only 12 and 16 free parameters for the Loire and the Durance catchments, respectively. Details about the parameters are given in Appendix A.

**104 2.2. Calibration and validation criteria**

We expect the model to provide a reliable hydrological behaviour for 105 the catchment. Therefore, it has to reproduce faithfully the various runoff 106 signatures, which reflect the different dynamics of its hydrology. The ob-107 served and simulated streamflows are then compared on the basis of four 108 numerical criteria. The Kling-Gupta Efficiency (KGE, Gupta et al. (2009)) 109 is calculated over four streamflow signatures: (i) the entire time-series (KGE 110 daily runoff), which is the result of all the processes, (ii) the inter-annual 111 daily regime (KGE daily regime), which reflects the interaction between wa-112 ter and energy availability as well as catchment storage, (iii) the average of 113 the monthly empirical cumulative distributions weighted by monthly runoff 114 (KGE flood), which focuses on floods produced by highly dynamic interac-115 tions and (iv) flow recessions during low flow period (KGE low flow), which 116 result from long-term processes (Blöschl et al., 2013; Garavaglia et al., 2017). 117 These four KGE criteria are used both for calibration and spatial validation. 118

For parameter calibration, the four criteria are implemented in the multiobjective genetic algorithm caRamel1 (Rouhier et al., 2017). Systematically, a first 1-year period is used for model spin-up. After 5000 runs, the algorithm provides a 4D Pareto frontier (Yapo et al., 1998) in which we select the set which minimises the Euclidian distance calculated on ranks.

**124 3. Data set**

**125 3.1. Study area**

The tailor-made method is assessed over two large French catchments: 126 the Loire basin at Gien (2a) and the Durance basin at Cadarache (2b). The 127 Loire catchment at Gien extending over  $35.707 \text{ km}^2$  is located in the central 128 part of France. Its elevation ranges from 118 to 1838 m.a.s.l. at which 129 the summits of the Massif Central peak. It is a mainly pluvial catchment 130 with a median elevation of 417 m.a.s.l. The Durance catchment at Cadarache 131 extending over 11,738 km2 is located in the Alps in south-east part of France. 132 Its elevation ranges from 247 to 4102 m.a.s.l. at which the Barre des Écrins 133 peaks. With 60% of the basin above 1000 m.a.s.l., the upper part is nival 134 while the lower part is nivo-pluvial. On top of that, the Durance catchment 135 is subject to karstic systems. In the south-west, a karstic formation supplies 136 the Fontaine d

---

## Author Comment (AC5) · 14 Feb 2019

[revised manuscript text omitted]

remote-sensing data. Along with streamflow observations, Khan et al. (2011) used satellite-derived flood maps to calibrate a module of a distributed model designed for flooding. Their results proved to be very promising for distributed hydrological model calibration, even in ungauged basins or data-sparse regions. Silvestro et al. (2015) then highlighted the usefulness of land-surface temperature and surface soil moisture satellite observations to reduce parameter equifinality. Their results also confirmed that the constraint method is a convenient alternative to calibrate a model in an ungauged context since they ended up with similar model performance for calibration over solely satellite-derived data and solely streamflow observations. More recently, Zink et al. (2018) also demonstrated the value of the land-surface temperature data for distributed modeling sine they obtained, over six German watersheds, very interesting runoff performance for a distributed model calibrated over solely land-surface temperature data. Regarding Demirel et al. (2018) and Wambura et al. (2018), they studied the value of the actual evapotranspiration derived from MODIS satellite observations. By simultaneously calibrating a distributed model over streamflow observations and spatial variability of evapotranspiration, they obtained a more consistent evapotranspiration estimation and similar or even better runoff performance, compared to a calibration over solely streamflow observations. They therefore demonstrated that this method provides a more robust parameter identification and substantially reduces equifinality and prediction uncertainty.

[revised manuscript text omitted]
. Indeed, according to preliminary attempts of regionalization not detailed here, we found out poor performance for validation stations whose drainage area ratio with the downstream calibration station is low. To address this, we propose to rearrange the calibration sub-basin pattern with physio-climatic information to become a physio-climatic calibration sub-basin pattern. To do so, the idea is to no longer inherit parameters from this calibration station but inherit parameters from the most similar calibration station in terms of physio-climatic descriptors. The selection of the new donor

[Figure]

**Figure 8.** Benchmark of the long-term mean monthly streamflow in terms of root-mean-square error. The physical method corresponds to the initial MORDOR-TS simulation with a single sub-basin pattern (see 4.1).

calibration station is carried out through a Euclidian distance calculated over the principal components of the physio-climatic descriptors. As basins are nested ones, applying this method for all the validation basins is not an option since it would implies the inter-dependance of all the parameter sets. Therefore, we adopt the 20% arbitrary value of drainage area ratio meaning that all the validation basins with a drainage area ratio lower than 20% are concerned by this physio-climatic transposition while
5   the parameters of the basins with a drainage area ratio higher than 20% continue to be transposed as before, *i.e.* as described in section 4.1. The new transposition patterns are presented in Fig. 4d and 5d. It is intended for all the remaining parameters about which we have no information or assumptions.

**5   Performance analysis**

The present paper does not claim to be exhaustive about the regionalisation methods. In fact, there are multiple possibilities
10  to regionalise a distributed hydrological model with transposition, prescription and constraint. Here, we propose to compare a single transposition with a combination of the three regionalisation methods. Indeed, the single transposition pattern presented in section 4.1 constitutes the initial reference in terms of the distributed parameters. It is referred to as *Exp1* and corresponds to the regionalisation method with the best overall performance of those discussed in citerouhier2017. The tailor-made pattern method developed in this paper consists in the four parameter patterns presented in section 4.2, namely a uniform pattern,
15  a hydrological mesh pattern, a sub-basin pattern and a physio-climatic calibration sub-basin pattern. This parameter scheme is referred to as *Exp2*. The two experiments are compared based on their performance over the four runoff signatures (daily

[revised manuscript text omitted]